# iSoMAs: Finding isoform expression and somatic mutation associations in human cancers

**Hua Tan**[1], **Valer Gotea**[1], **Sushil K. Jaiswal**[1], **Nancy E. Seidel**[1], **David O. Holland**[1], **Kevin Fedkenheuer**[1], **Abdel G. Elkahloun**[2], **Sara R. Bang-Christensen**[1], **Laura Elnitski**[1,*]

**1** Translational and Functional Genomics Branch, National Human Genome Research Institute, National Institutes of Health, Bethesda, Maryland, United States of America, **2** Microarrays and Single-Cell Genomics Core, National Human Genome Research Institute, National Institutes of Health, Bethesda, Maryland, United States of America

\* elnitski@nih.gov

## Abstract

Aberrant alternative splicing, prevalent in cancer, impacts various cancer hallmarks involving proliferation, angiogenesis, and invasion. Splicing disruption often results from somatic point mutations rewiring functional pathways to support cancer cell survival. We introduce iSoMAs (**iSo**form expression and somatic **M**utation **As**sociation), an efficient computational pipeline leveraging principal component analysis technique, to explore how somatic mutations influence transcriptome-wide gene expression at the isoform level. Applying iSoMAs to 33 cancer types comprising 9,738 tumor samples in The Cancer Genome Atlas, we identified 908 somatically mutated genes significantly associated with altered isoform expression across three or more cancer types. Mutations linked to differential isoform expression occurred through both *cis-* and *trans-*acting mechanisms, involving well-known oncogenes/suppressor genes, RNA binding protein and splicing factor genes. With wet-lab experiments, we verified direct association between *TP53* mutations and differential isoform expression in cell cycle genes. Additional iSoMAs genes have been validated in the literature with independent cohorts and/or methods. Despite the complexity of cancer, iSoMAs attains computational efficiency via dimension reduction strategy and reveals critical associations between regulatory factors and transcriptional landscapes.

## Author summary

Somatic single nucleotide variants (SNVs) drive human cancer progression by disrupting alternative splicing (AS), a co-transcriptional process that generates transcript variation and proteome diversity. To better understand the regulatory networks governing splicing, we propose a computational pipeline, iSoMAs (**iSo**form expression and somatic **M**utation **As**sociation), and systematically investigate associations between SNVs and isoform expression in a pan-cancer analysis. Our approach leverages principal component analysis (PCA) for dimension reduction in sample-matched mutation and expression data from The Cancer Genome Atlas (TCGA). We identify thousands of

**Data availability statement:** Data generated in this study are published with this work as supplementary materials. We also created a GitHub repository (https://github.com/elnitskilab/iSoMAs) for this project, which includes all R codes, tutorials and demo datasets that demonstrate usage of the iSoMAs pipeline and interpretation of its output.

**Funding:** This work was supported by the Intramural program of the National Human Genome Research Institute, National Institutes of Health (ZIAHG200323 to LE) and the Carlsberg Foundation (CF21-0592 to SRB-C). The funders had no role in study design, data collection and analysis, decision to publish, or preparation of the manuscript.

**Competing interests:** The authors have declared that no competing interests exist.

genes (termed iSoMAs genes) whose mutation is significantly associated with isoform expression of multiple genes located across different chromosomes in 33 cancer types. Prevalent iSoMAs genes include the well-characterized tumor suppressor *TP53* and splicing factor *SF3B1*. iSoMAs outperforms traditional association study methods with high computational efficiency and stringent control of false positives. Our results show strong biological and clinical relevance, reflecting known and novel functional relationships. We further demonstrate the *TP53*, R273H mutation's ability to alter isoform expression by reversing its effects in lung cancer cells. Our findings bring insights into the regulatory networks of isoform splicing and transcription in cancer, bridging genetic and epigenetic regulation of human oncogenesis in an innovative and biologically meaningful way.

## Introduction

Alternative splicing (AS) is a prevalent biological process responsible for the majority of transcript structural variation and proteome diversity within human genomes [1]. The expression of alternative isoform transcripts contributes functionality to the 'hallmarks of cancer' by driving tumor growth and metastatic progression [2–4]. Somatic disease-causing mutations (i.e., driver mutations) in both coding and noncoding regions [5] can exert their functions by disrupting gene transcription and splicing processes through a variety of *cis*- and *trans*-regulatory mechanisms. Specifically, somatic mutations can disrupt local gene splicing through mechanisms such as exon skipping, intron retention or frameshifts (*cis*-effects) [6–9]. They can also impact gene splicing in distal locations if occurring on their regulatory RNA binding protein (RBP), splicing factor (SF) or transcription factor (TF) genes (*trans*-effects) [10–12]. These factors are responsible for balancing isoform expression through regulated alternative splicing in healthy cells. However, the complexity of cancer indicates that the functional imbalances cannot be attributed to a single mutation, an alternatively spliced exon, or a differentially expressed isoform. Rather, they arise from myriad combinations that describe the cancer cell as a system.

Numerous prior studies have underscored the important role of somatic mutations, particularly single nucleotide variants (SNVs), in AS in human cancers. For example, a study of 1,812 patients from six cancer types using TCGA (The Cancer Genome Atlas) database identified ~900 somatic splicing mutations in coding regions, of which 163 SNVs are likely to cause intron retention or exon skipping [7]. SNVs from this work demonstrated enrichment in tumor suppressor genes (TSGs), confirming a common mechanism of TSG inactivation through splicing alternation. Another study known as MiSplice (mutation-induced splicing) analyzed ≥8,000 tumor samples across all 33 TCGA cancer types. The authors characterized 1,964 somatic mutations with evidence of creating alternative splice junctions within ±20 bp from each mutation, termed "splice-site-creating mutations" (SCMs), indicating *cis*-effects [13]. These SCMs are biologically important as they generate neoantigens with greater immunogenicity than regular missense mutations, which makes them good candidates for immune therapy. A third study conducted a pan-cancer association analysis of genetic variation and AS referred to as sQTL (splicing quantitative trait loci), using genomics data from the same 33 TCGA cancer types. This study identified 32 *cis*- and 7 *trans*-acting sQTLs, including those on the well-known splicing factors *SF3B1* and *U2AF* [14]. Finally, a series of other studies focused on somatic mutations on known SFs and confirmed the *trans*-acting effects of somatic SNVs on gene splicing and consequent clinical outcomes, using both computational and experimental approaches [15–17]. Together, these studies have firmly established that somatic SNVs that disrupt gene splicing represent a major pathogenic mechanism in human cancers.

Despite many merits of these existing studies, methodological limitations may explain the omission of many sQTLs and SNV-associated AS events. In particular, the existing pipelines and models are invariably limited to a direct, one-to-one mapping between an SNV and a specific, mostly local AS event (e.g., measured by the level of percent spliced in (PSI), or the presence of exon skipping/intron retention), producing gene-by-gene associations. Specifically, in their search for *cis*-sQTLs, previous studies restricted their investigation to the flanking exons or neighboring exon-intron junctions. Similarly, when searching for *trans*-sQTLs, existing studies considered only known SF and splicing regulatory genes. These strategies excluded the exploration of SNVs occurring in many other RBPs and co-regulators, especially those acting upstream of splicing machinery (e.g., TFs targeting the splicing machinery) that work in an indirect way. A striking example of indirect regulation is observed in the effects of mutant *TP53* on RNA splicing, achieved through the upregulation of an intermediate RBP called hnRNPK [18]. In summary, the direct gene-by-gene association method has two methodological drawbacks: (i) it necessitates a huge number of SNV-AS pairs in order to exhaust all possible combinations, incurring an intense computational burden, especially for a pan-cancer analyses; and (ii) the statistical power is diminished due to the generally low mutation frequency of most cancer driver genes [19,20], which challenges the multiple testing correction process to reduce false positive rate and accommodate different cancer types.

To address these challenges and methodological gaps, we developed a novel computational pipeline called iSoMAs (**iSo**form expression and somatic **M**utation **As**sociation) that integrates matched DNA-seq and RNA-seq data of each cancer type to efficiently identify the associations between somatic SNVs and isoform expression profiles, taking advantage of the principal component analysis (PCA) technique. Briefly, instead of examining one-to-one association between each SNV and the expression of each transcript (isoform) directly, iSoMAs evaluates the association between each inquired SNV and the overall expression pattern of the entire transcriptome consisting of ~73K annotated transcripts, followed by assignment of the association to each of the constituent isoforms based on the PCA decomposition. Applying iSoMAs to 33 TCGA cancer types separately, we detected 908 iSoMAs genes (i.e., genes eventually selected by the iSoMAs pipeline) present in three or more cancer types. We also detected additional iSoMAs genes that are specific to one or two cancer types. These iSoMAs genes include many well-recognized oncogenes, tumor suppressors and RBPs/SFs/TFs, indicating their biological and clinical relevance. Moreover, numerous iSoMAs genes have been validated for their impacts on splicing of predicted targets by prior studies using independent cohorts and/or methods. We also performed independent validation of isoforms altered in the presence of *TP53* mutations in lung cancer cells, by reversing the effect of the *TP53* mutations with a small molecule. We emphasize that iSoMAs is the first dimension reduction-based method to systematically explore the association between somatic SNVs and the transcriptome-wide isoform expression across pan-cancer. Meanwhile, iSoMAs offers computational efficiency without compromising statistical rigor. It comprehensively models the systems biology of cancer cells by elucidating networks encompassing direct and indirect regulation of alternative splicing and transcription.

## Results

### Gene isoform expression and somatic mutation profiles provide new molecular taxonomy beyond histopathological classification

To study the association between gene mutation and gene isoform expression, we examined matched RNA-seq and DNA-seq (whole-exome) data from The Cancer Genome Atlas (TCGA) (Fig 1A). The RNA-seq data encompassed the expression levels of 73,599 isoforms

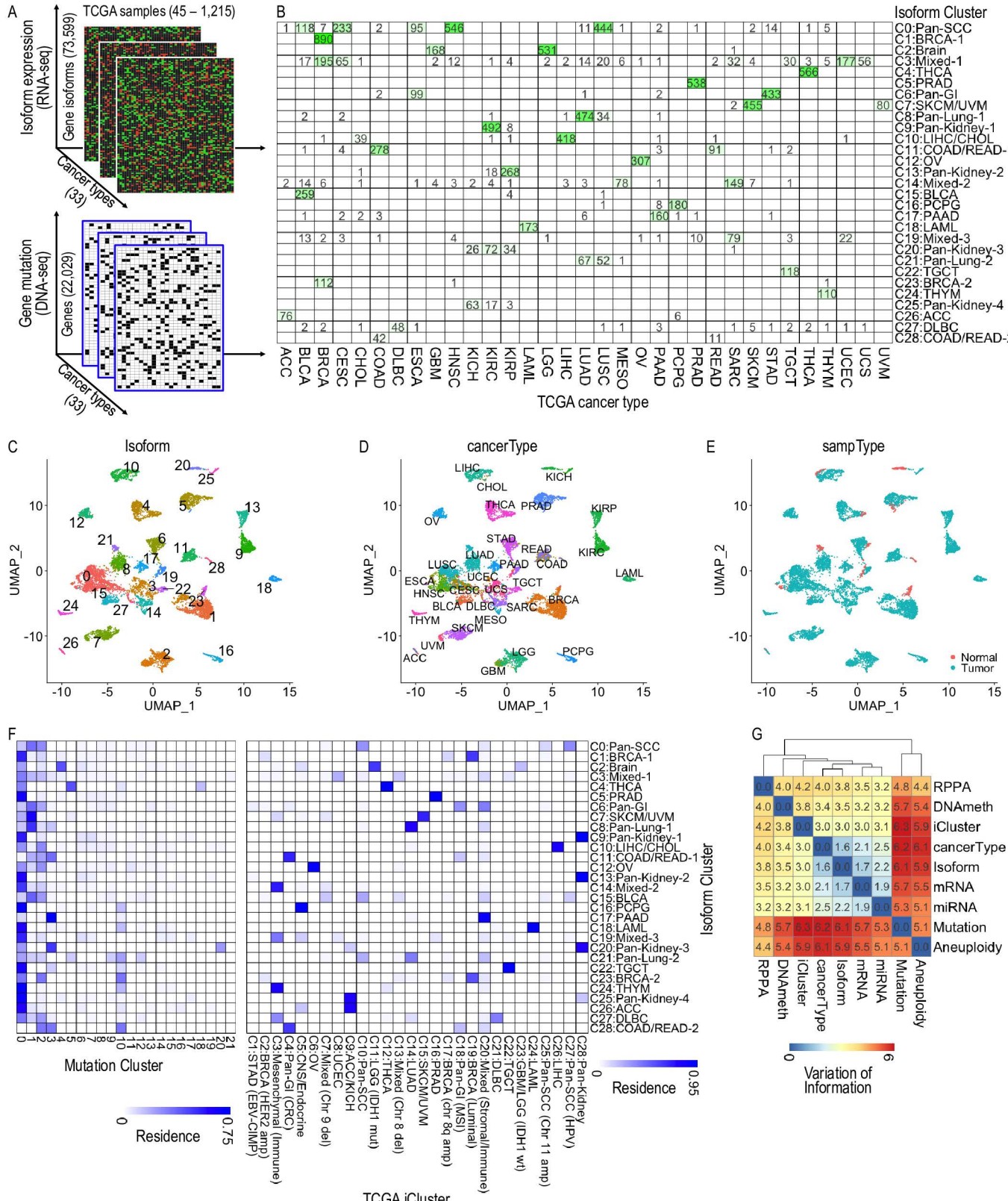

**Fig 1. Gene isoform expression and somatic mutation profiles provide new molecular taxonomy beyond tissue-histology classifications.** (A) TCGA multi-omics data employed in the study include matched RNA-seq for isoform expression (upper) and whole-exome-wide DNA-seq data for gene somatic

mutation (lower) across 33 cancer types. (B) Cluster residence heatmap shows the number of samples from a given cancer type that resides within each of the 29 (C0 – C28) annotated Isoform clusters. (C) The UMAP visualization of 10,464 TCGA cancer samples based on expression of 73,599 isoforms. Each dot represents a TCGA sample and is colored/marked by the 29 Isoform clusters. (D-E) Samples on the same Isoform UMAP map are colored by TCGA cancer types (D) and tumor status (E). (F) Cluster residence heatmap shows the percent of each Isoform cluster that overlaps with each Mutation cluster (left) and TCGA iCluster (right). (G) Variation of information analysis of clustering schemes derived from various TCGA data types. The cluster membership of Aneuploidy, mRNA, miRNA, RPPA, DNA methylation (DNAmeth) and iCluster were derived from (Hoadley et al., 2018). The membership of Isoform was determined in this study. See also S1 Table and S1 Fig.

derived from 29,181 genes, comprising both coding (20,429) and noncoding (8,752) genes (S1 Table). The DNA-seq data detected somatic mutations in 22,029 genes. These data were sourced from a total of 10,464 cancer samples, including 9,738 tumor samples and 726 tumor-adjacent normal tissue samples spanning 33 cancer types (45 - 1,215 samples per cancer type).

Clustering analysis based on gene isoform expression (measured as Fragments Per Kilobase of transcript per Million mapped reads, or FPKM, by the RSEM algorithm, see Methods) across all 33 cancer types (see Methods) identified 29 distinct "Isoform" clusters (labeled as C0–C28, Fig 1B). We annotated these clusters according to the observed enrichment in specific cancer types. For instance, cluster C2 was labeled as "Brain" since most samples in this cluster originated in brain tissue-related cancers: GBM (Glioblastoma) and LGG (Lower-grade Glioma). The 29 molecular subtypes generally well reflected the tissue origin of samples in each cluster (Fig 1B-D). We further noticed that the isoform expression profile clearly distinguished the tumor and normal samples (Fig 1E). In addition, several intriguing patterns were observed from the clustering results. For example, samples from the same or adjacent organ/tissue tended to cluster together, such as the Pan-GI (gastrointestinal) cluster (C6 ~ ESCA+STAD), the COAD/READ (large intestine) clusters (C11 and C28 ~ COAD+READ), the Pan-Lung clusters (C8 and C21 ~ LUAD+LUSC), and the Pan-Kidney clusters (C9 and C13 ~ KIRC+KIRP, C20 and C25 ~ KICH+KIRC+KIRP). Additionally, tumors originating from the same cell type were likely to show similar isoform expression patterns, such as the Pan-SCC (squamous cells) cluster (C0 ~ CESC+HNSC+LUSC) and SKCM/UVM (pigment-producing cells) cluster (C7 ~ SKCM+UVM). Some cancer types split into two or more distinct clusters such as the breast cancer (BRCA ~ C1+C23+C3) and esophageal cancer (ESCA ~ C0+C6) cancers. We also detected three mixed clusters containing samples from multiple cancer types (C3, C14 and C19). It's worth noting that the subclusters of BRCA samples differ from the five classical BRCA subtypes derived from PAM50 mRNA profiles [21] (S1A Fig). This difference can be attributed to the clustering being performed across multiple cancer tissue types, where tissue-specific differences may dominate over subtype specific differences within the same tissue.

Clustering analysis of the same TCGA tumor samples based on whole-exome somatic mutation data (see Methods) detected 22 "Mutation" clusters (labeled as 0–21, S1B Fig). Compared to gene isoform expression, somatic single nucleotide variant (SNV) profiles exhibited much diminished power of discrimination between cancer types, as indicated by the mixed composition of almost all the 22 clusters (S1B-C Fig). The cluster residence map (Fig 1F, left) better illustrates this observation with the nontrivial overlap between one Mutation cluster and multiple Isoform clusters, especially Mutation clusters 0-3. However, we found strong concordance between our isoform expression-based clustering results and a multiomic features-based clustering ("iCluster") results [22] (Fig 1F, right). The iCluster scheme incorporates four TCGA data types, including copy number (aneuploidy), DNA methylation, mRNA and microRNA (miRNA), to partition all the TCGA tumor samples into 28 clusters (S1D Fig). Comparing to other TCGA-based clustering schemes with variation of information

(VoI) analysis (see Methods), we observed the strongest concordance of our Isoform clustering scheme with cancer type and mRNA, followed by miRNA and iCluster (Fig 1G). The VoI analysis further shows that the gene isoform expression provides higher resolution of cancer type classification compared to mRNA at the gene level (1.6 vs. 2.1 in VoI). This finding aligns with our previous discovery that the differential isoform ratios exhibit minimal correlation with the differential expression at gene level in the human kinome in SKCM cancer [4].

Taken together, gene isoform expression and gene somatic mutation profiles provide unique molecular taxonomy in addition to the organ/tissue histology-based pathology classification. This extends beyond other phenotypic characteristics such as tumor stage and tumor tissue purity (S1E Fig). While isoform and mutation-based molecular signatures stratify cancer samples at different resolutions, integrating their information content might better characterize the intrinsic molecular traits of tumor samples. Therefore, the sample-matched RNA-seq and DNA-seq data across multiple cancer types represent a great opportunity to investigate the association between somatic gene mutations and isoform expression patterns.

## iSoMAs efficiently detects somatic mutations associated with gene isoform expression

Utilizing the matched TCGA multi-omics data as described above, we devised the iSoMAs pipeline to investigate the potential association between gene somatic mutation and gene isoform expression (Fig 2A). The iSoMAs workflow consists of two steps: In the first step, the high-dimensional gene isoform expression matrix (d=59,866 derived from the 15,448 multi-isoform genes) for each cancer type is trimmed by the mean.var.plot method (built in the Seurat toolkit [23], see Methods) into a more informative expression matrix. This step keeps only the most variable isoforms but still maintains high-dimensionality (d≈3,500). Afterward, the Principal Component Analysis (PCA) [24] is performed to further reduce the dimension of the informative expression matrix into a much lower-dimensional PC score matrix (d=50) by calculating a PC loading matrix [25]. Each column of the PC loading matrix corresponds to a particular linear combination of the input isoforms, resulting in a meta-isoform, with the coefficients of the combination stored in the respective column of the PC loading matrix. The coordinates of the new low-dimensional expression space are formed by all the meta-isoforms.

In the second step, a differential PC score analysis is conducted along each of the 50 PC coordinates based on the mutation status (SNV) of the studied gene, using Wilcoxon rank-sum test. Following the differential PC score analysis for each test gene, an iSoMAs gene is determined if (1) the minimum of the Bonferroni corrected p-values ($P^{adj}$) along all 50 PC coordinates is smaller than a predefined threshold ($P^{adj}_{min} < 0.05$, or equivalently the original $P_{min} < 0.001$ for 50 tests) and (2) the $P_{min}$ of the candidate gene passes a secondary multiple testing correction with FDR<0.05. It should be noted that, a single gene may exhibit significant associations with meta-isoforms along multiple PC coordinates (referred to as significant PCs hereafter). In such cases, we select the first significant PC (i.e., the one with smallest index among all significant PCs) for downstream analysis unless otherwise stated, since in PCA the first PCs explain much greater proportion of variance compared to their succeeding ones. In our practice, the first significant PC turned out to be the most significant PC for most iSoMAs genes in most cancer types, i.e., $P_{first} = P_{min}$ (S2B Table). Along this selected PC we resort to the PC loading matrix to prioritize (rank) all input transcripts (which typically include both *cis*- and *trans*-positions) that are potentially significantly associated with this SNV, and hence establish the association between this SNV and the expression of those top-ranked transcripts (isoforms).

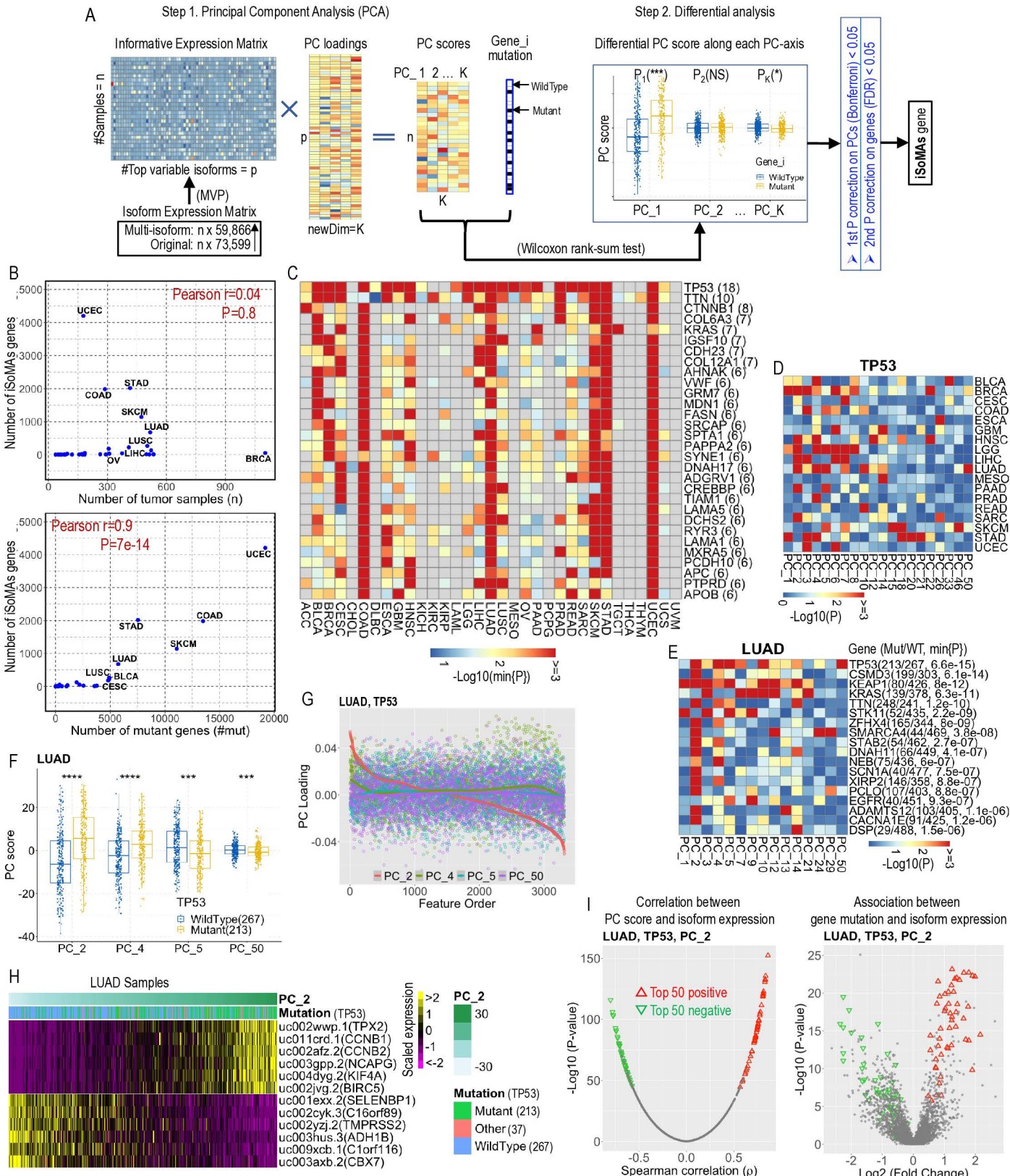

**Fig 2. Overview of the iSoMAs workflow and iSoMAs genes across pan-cancer.** (A) The iSoMAs workflow consists of two steps: dimension reduction with PCA followed by a differential PC score analysis with Wilcoxon's rank-sum test. An iSoMAs gene is determined if it passes two-layer statistical significance test, i.e.,

at both the PC and mutant gene levels. The MVP method (developed in Seurat) was used to trim the original isoform expression matrix into a more informative expression matrix which keeps the top variable isoforms only. (B) The number of iSoMAs genes detected in each cancer against the sample size (upper) and the number of mutant genes (lower). (C) Heatmap shows the Log10-transformed minimum p-value along the 50 PC coordinates for the top 30 iSoMAs genes across pan-cancer. Genes were ranked by the number of cancer types in which they were detected as iSoMAs genes, as indicated in parentheses. (D) TP53 was detected as an iSoMAs gene in 18 cancer types, involving 20 PC axes as indicated. (E) The top 18 iSoMAs genes detected in LUAD cancer. Heatmap shows the significance level of each gene across the 50 PC axes. Number of wildtype (WT) and mutant (MUT) samples, and the minimum p-value among the 50 PCs for each iSoMAs gene are indicated. PCs not significant in any of the 18 genes in LUAD were excluded for clarity. (F) TP53 was tested significant along four PC axes: PC_2, PC_4, PC_5 and PC_50 in differential PC score analysis for LUAD. ****$P$<1e-4, ***$P$<1e-3 by Wilcoxon rank-sum test. (G) Visualization of the PC loadings along significant PC axes for TP53 in LUAD cancer. Isoforms (x-axis) were ordered by PC loadings along PC_2 axis. (H) Scaled expression levels of the top 6 positive (rows 1-6) and top 6 negative (rows 7-12) isoform targets of TP53 for all LUAD tumor samples ordered by their PC scores along PC_2 axis. The isoforms were ranked by the PC loadings along PC_2 axis. (I) Left: Spearman correlation between PC_2 score and expression level of each of the input 3,315 isoforms across 517 LUAD tumor samples. Right: Association between TP53 mutation and expression level of each of the input 3,315 isoforms across 517 LUAD tumor samples, measured by two-sided Wilcoxon rank-sum test. The top 50 positive and top 50 negative isoforms (ranked by PC_2 loadings) are marked red and green, respectively. See also S2 Table and S2 Fig.

We applied iSoMAs to 33 TCGA cancer types individually and detected varied numbers of iSoMAs genes (Table 1) across different cancers. Specifically, we identified more than 2,000 iSoMAs genes in STAD and UCEC cancers which are known for hypermutability, but none in CHOL and KICH cancers. We obtained only one iSoMAs gene in three cancer types: DLBC (*PIM1*), LAML (*RUNX1*) and UCS (*FBXW7*). Considering the top 3 most significant iSoMAs genes (ranked by $P_{\min}$) detected in each cancer type, some are shared by multiple cancers (e.g., *TP53* are shared by 12 cancers; the RAS family including *KRAS*, *HRAS* and *NRAS* are shared by 5 cancers). The number of iSoMAs genes detected in each cancer type showed little correlation with the sample size (Pearson r=0.04, $P$=0.8), but showed significant correlation with the number of mutant genes detected in that cancer (Pearson r=0.9, $P$=7e-14), especially for cancers with more than 5,000 mutant genes (Fig 2B).

We aggregated these genes to create a master list of 7,140 unique (i.e., non-duplicated) iSoMAs genes and ranked them based on the number of cancer types in which they were detected as an iSoMAs gene (cancer coverage) (S2A-B Table), ranging from 1 to 18. Specifically, out of the 7,140 iSoMAs genes, 2,558 (36%) were present in two or more cancer types, while the remaining 4,582 genes were found exclusively in one cancer type. Many well-known oncogenes (e.g., *CTNNB1* and the RAS family), tumor suppressors (e.g., *TP53* and *PTEN*) and RNA-binding protein genes (e.g., *RBM10*, *TNRC6A* and *SF3B1*) were detected as iSoMAs genes in multiple cancers, implicating their involvement in isoform production across various pathological contexts.

Each of the top 30 iSoMAs genes, ranked by cancer coverage (S2A Table), was present in six or more cancer types. Additionally, 17 out of the 30 genes were shared by the top four cancer types (UCEC, STAD, COAD and SKCM) with the most iSoMAs genes (Fig 2C, S2A Table). *TP53* turned out to be the most prevalent iSoMAs gene, being present in 18 out of 33 TCGA cancer types (Fig 2D).

As the iSoMAs method utilizes PCA, we examined the biological relevance of each PC axis. We found that the iSoMAs genes could exhibit significance along any of the 50 PC axes, as evidenced by the nontrivial percentages represented by the last 10 PCs (PC_41 – PC_50) (Figs 2D-F and S2A). We further confirmed that the PC loading values along different PC axes were quite different among each other (Fig 2G), which is reasonable in that PCA tries to capture complementary information inherent in the input matrix through different coordinates in the transformed space. In addition, we mathematically proved that the significant association between a gene mutation and a PC score is equivalent to the significant association between the gene mutation and the expression level of isoforms with large (either positive or negative) PC loading values, owing to the linear combination property of the PCA decomposition (see Methods). This equivalence is further illustrated by the *TP53* example in the LUAD cancer

**Table 1. A summary of the iSoMAs analysis for 33 TCGA cancer types.**

| Cancer | n | p | Test | iSoMAs | Gene#1 | Gene#2 | Gene#3 |
|---|---|---|---|---|---|---|---|
| ACC | 79 | 4007 | 121 | **2** | MUC16 | CTNNB1 | |
| BLCA | 408 | 3687 | 4886 | **218** | TP53 | FGFR3 | RB1 |
| BRCA | 1102 | 3271 | 377 | **48** | TP53 | PIK3CA | CDH1 |
| CESC | 306 | 3418 | 4817 | **179** | EP300 | C6 | MYH15 |
| CHOL | 36 | 3647 | 10 | **0** | | | |
| COAD | 287 | 3391 | 13491 | **1981** | BRAF | MYO18A | HERC2 |
| DLBC | 48 | 3861 | 69 | **1** | PIM1 | | |
| ESCA | 185 | 4009 | 2191 | **50** | LAMA1 | PREX2 | NFE2L2 |
| GBM | 169 | 3742 | 3571 | **12** | TP53 | IDH1 | SLC34A3 |
| HNSC | 522 | 3705 | 1927 | **125** | TP53 | NSD1 | NFE2L2 |
| KICH | 66 | 3718 | 3 | **0** | | | |
| KIRC | 534 | 3246 | 50 | **6** | PBRM1 | BAP1 | MTOR |
| KIRP | 291 | 3935 | 281 | **5** | MET | MUC2 | SLC26A5 |
| LAML | 173 | 3502 | 18 | **1** | RUNX1 | | |
| LGG | 534 | 3810 | 161 | **10** | IDH1 | TP53 | EGFR |
| LIHC | 374 | 3400 | 1087 | **36** | CTNNB1 | TP53 | KEAP1 |
| LUAD | 517 | 3315 | 5725 | **677** | TP53 | CSMD3 | KEAP1 |
| LUSC | 502 | 3393 | 4896 | **266** | NFE2L2 | KEAP1 | ZIM2 |
| MESO | 87 | 3276 | 9 | **3** | TP53 | LATS2 | NF2 |
| OV | 309 | 3745 | 2550 | **24** | FLNB | ZNF835 | PIK3CA |
| PAAD | 179 | 3507 | 1008 | **4** | KRAS | TP53 | PEG3 |
| PCPG | 184 | 4191 | 7 | **2** | HRAS | EPAS1 | |
| PRAD | 498 | 3553 | 134 | **10** | SPOP | TP53 | MXRA5 |
| READ | 95 | 3064 | 3768 | **22** | APC | PEG3 | MED12L |
| SARC | 263 | 3651 | 575 | **9** | TP53 | MKI67 | FLT4 |
| SKCM | 472 | 3343 | 11087 | **1140** | CRB1 | DNAH7 | TTN |
| STAD | 415 | 3609 | 7526 | **2013** | PLEC | PIK3CA | MROH1 |
| TGCT | 156 | 3877 | 7 | **2** | KIT | KRAS | |
| THCA | 513 | 3437 | 6 | **3** | BRAF | NRAS | HRAS |
| THYM | 120 | 3852 | 8 | **2** | GTF2I | HRAS | |
| UCEC | 177 | 3827 | 19162 | **4205** | TP53 | MYH3 | APOB |
| UCS | 57 | 3893 | 115 | **1** | FBXW7 | | |
| UVM | 80 | 3387 | 5 | **2** | SF3B1 | EIF1AX | |

n: number of samples; p: number of top variable isoforms as input (see Fig 2A); Test: number of mutant genes tested; iSoMAs: number of iSoMAs genes detected; Gene#1-3: the top 3 iSoMAs genes detected in each cancer type, ranked by the minimum p-value derived in the iSoMAs analysis.

(Figs 2H-I and S2B-E; S2C Table) and a thorough validation and benchmarking analysis for all iSoMAs genes detected in all 33 TCGA cancer types (S2G-J Table). In addition, as the importance of PCs decreases with index (S2F Fig), measured by the proportion of total variance explained (S2G Fig), this equivalence becomes much more notable at the lower-index PC axis compared to the higher-index ones (Figs 2H-I; S2H-I; S2G Table).

To address the relevance of alternative splicing in these associations, we further checked the special cases where the mutation status of the iSoMAs gene is significantly associated with the altered expression of a target gene's isoforms in dual directions. Specifically, an iSoMAs gene mutation could significantly (Wilcoxon's $P<0.05$) upregulate some isoforms while downregulating the other isoforms of the same target gene. In total, we found that 4,986 (i.e., 69.8% of

all 7,140) iSoMAs genes exhibited dual-direction associations with at least one gene (pooled mean: 2; range: 1-68) representing their top 100 target isoforms (S2J-L Fig, S2D Table). Interestingly, *TP53* showed highest significance level in dual-direction association in six cancer types, including BRCA, COAD, GBM, LUAD, PRAD and SARC (S2L Fig). It should be noted that these isoform switches occur between mutant and wildtype tumor samples regarding some iSoMAs genes, rather than between tumor and normal samples as previously studied [26]. However, we did observe that 77 out of the 233 (33%) tumor-normal switching genes (involving 244 isoform transcripts) overlapped with our iSoMAs dual-direction associations (appeared in the iSoMAs targets). These data demonstrate that altered isoform expression associated with a gene mutation must involve alternative splicing because the exon content of the isoforms changed simultaneously with the divergence in isoform expression levels.

Together, these results clearly demonstrate the effectiveness and efficiency of our proposed iSoMAs pipeline in (1) identifying genes (i.e., iSoMAs genes) whose mutation status is significantly associated with the isoform expression at the transcriptome level in each cancer type and (2) prioritizing the cross-chromosome isoforms (i.e., iSoMAs targets) whose expression is potentially impacted by the mutant gene as queried. This dimension reduction-based approach utilizing PCA proved much more efficient than the traditional differential expression analysis applied directly to the original isoform expression.

## iSoMAs genes tend to associate with gene isoform expression in a *trans*-regulatory manner

We further explored the iSoMAs genes and their associated gene isoforms (transcripts) having the largest PC loading weights determined in each cancer type. Those top 100 isoforms (unless otherwise stated) ranked by the PC loading value are termed the target isoforms of this iSoMAs gene in the subsequent analysis. It deserves noting that the number of top isoforms selected for downstream analysis is not biologically critical, we started with an initial number suitable for a meaningful downstream analysis such as enrichment profiling. Also, the PC loadings typically decay in an exponential-like pattern over index and show an obvious turning point around 100 (Fig 2G). We clarify that while there might be more or less than 100 isoforms really associated with the same iSoMAs gene, which subjects to validation (e.g., Figs 2I, S2D-E), to maintain consistency throughout all downstream analyses and across different cancer types, we truncated the lists at 100 members.

We first examined the genomic distribution of the top isoform targets of the iSoMAs genes in each cancer type. In all 31 TCGA cancer types with iSoMAs genes detected, the top 24 isoform targets of an iSoMAs gene were typically distributed across 10-15 chromosomes in any cancer type (median = 12 with all cancers pooled). Furthermore, the top 100 isoform targets of each iSoMAs gene typically spanned 20 or more chromosomes (pooled median = 21) (Fig 3A). This observation indicates that an iSoMAs gene could potentially impact the gene isoform expression in a genome-wide manner.

Next, we checked whether there is any chromosome preference for the isoform targets of an iSoMAs gene in each cancer type. To do this, for each iSoMAs gene detected in each cancer type, we compared the frequency distribution of the top 100 isoform targets along the 24 chromosomes (including X and Y) to the number distribution of all isoforms (Niso) across the 24 chromosomes. We calculated a p-value for this comparison with the Kolmogorov-Smirnov (KS) test. We observed few iSoMAs genes that passed the KS test at the P=0.05 (or -Log10(P)=1.3) level (Figs 3B and S3A), indicating that few of them showed significantly different probability distributions. This means that the isoform targets of most iSoMAs genes in any cancer type have little chromosome preference.

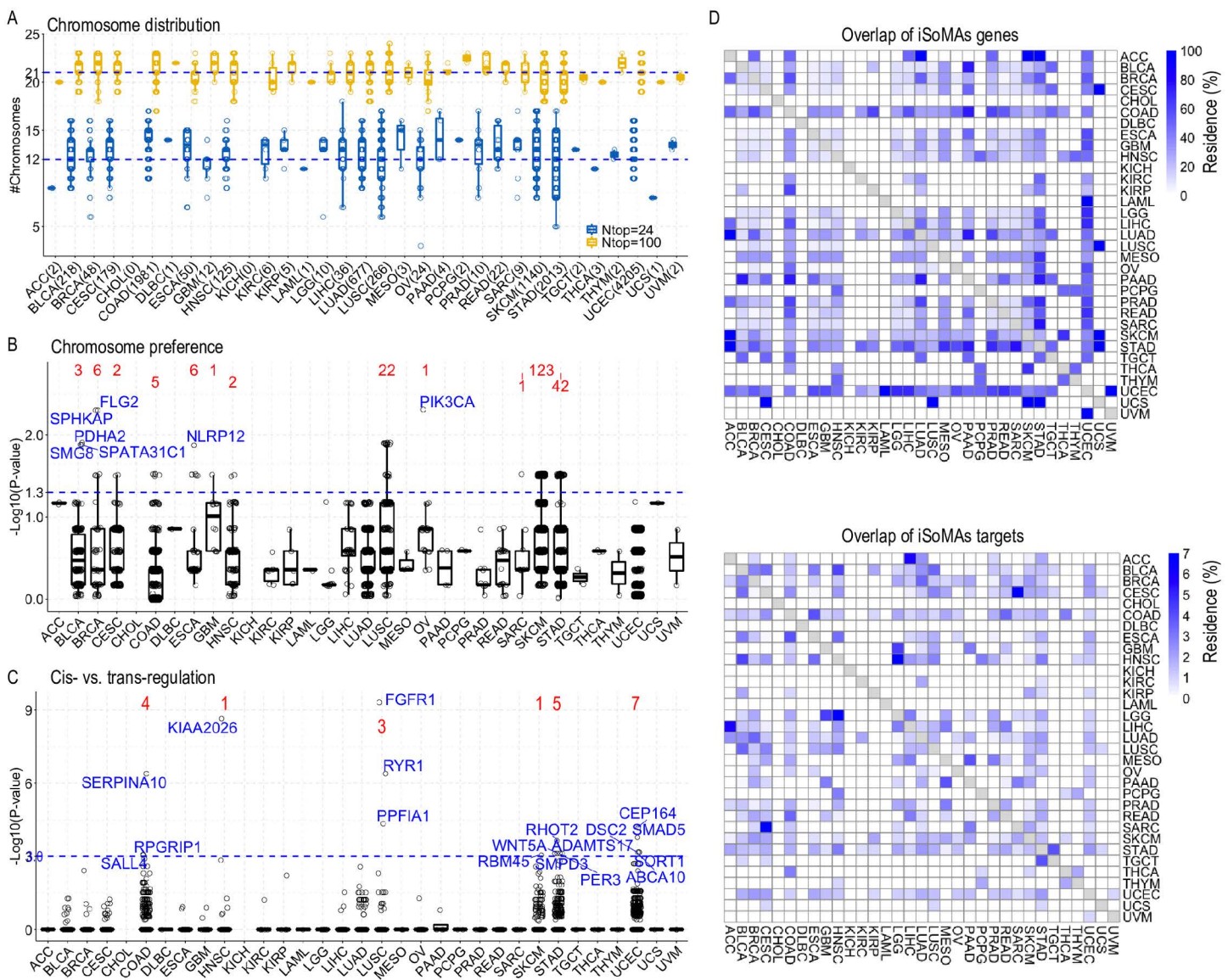

**Fig 3. *Cis-* and *trans-*regulation patterns of iSoMAs genes across pan-cancer.** (A) Chromosome distribution of target isoforms of the iSoMAs genes detected in each cancer type. Y-axis represents the number of chromosomes across which the top 24 and 100 target isoforms are distributed. Dotted lines indicate the pooled median for all cancer types. Numbers in parentheses in the x-axis labels indicate the number of iSoMAs genes detected in each cancer (Table 1). (B) Chromosome preference of target isoforms of the iSoMAs genes. For each iSoMAs gene detected in each cancer type, the frequency distribution of its top 100 target isoforms along 24 chromosomes was compared to the number distribution of all isoforms across the 24 chromosomes by KS test. Y-axis represents the Log10-transformed p-value derived from the KS test. The number above each box whisker indicates the number of iSoMAs genes yielding P<0.05 in the KS test in corresponding cancer type. (C) Boxplot illustrates the likelihood that iSoMAs genes target isoforms in a *cis*-acting mechanism. Y-axis represents the Log10-transformed p-value derived from the hypergeometric test measuring the likelihood that the top 100 target isoforms of an iSoMAs gene are in its *cis*-regulation region (±1Mb of TSS). Numbers of iSoMAs genes with P<0.001 in the hypergeometric test and the most significant iSoMAs gene are indicated above corresponding cancer types. (D) Pairwise overlap of iSoMAs genes (upper) and iSoMAs targets (lower) among cancer types. The top 100 isoform targets of each iSoMAs gene were extracted and pooled together for each cancer type prior to calculating the pairwise overlap among 33 cancer types. The percent of overlap (Residence) between two iSoMAs gene sets was calculated as the size of overlap divided by the size of the smaller set. See also S3 Table and S3 Fig.

We further asked whether iSoMAs genes tend to target their neighboring genes (or *cis*-region, defined as ±1Mb of its TSS) compared to distal genes (or *trans*-region, defined as non-*cis*-regions). To address this question, we conducted a hypergeometric test to measure the significance level of a particular number (or more) of the top 100 isoform targets were

located in the *cis*-region of a designated iSoMAs gene (S3B Fig). We observed that very few iSoMAs genes passed the hypergeometric test for *cis*-regulatory iSoMAs genes in all cancer types at the *P*=0.001 level (Fig 3C; S3A-B Table). The number of *cis*-regulatory iSoMAs genes increased a little at the *P*=0.05 and *P*=0.01 levels but remained small (typically less than 5%). In very special cases, iSoMAs genes can target themselves for differential isoform expression, e.g., *FGFR3* targets different isoforms of itself in BLCA and KIRP cancers (S3C Table). While it's hard to claim that all the remaining genes are *trans*-regulatory iSoMAs genes, those with a hypergeometric test *P*>0.999 (indicating a p-value<0.001 for the test with the opposite null hypothesis) represented the majority (S3C Fig). These results indicate that very few iSoMAs genes detected in each cancer type prefer to target genes in *cis*-locations compared to *trans*-locations.

While the majority (64.2%) of iSoMAs genes were specific to only one cancer type (S2A Table, we found that the whole iSoMAs gene set of many cancer types largely overlapped with that of other cancer types (Fig 3D, upper). These inter-cancer overlapping patterns of iSoMAs genes generally extended to their isoform targets in most cancers, although the scale of overlapping was obviously diminished (Fig 3D, lower). These results demonstrate an underlying commonality in the regulation of at least some alternative splicing events associated with somatic mutations across various cancer types.

Looking into the specific [iSoMAs gene]:[iSoMAs target] pairs, we observed distinct patterns in inter-cancer overlap profiles compared to that with iSoMAs genes or iSoMAs targets alone. While apparent pair-wise overlap in [iSoMAs gene]:[iSoMAs target] pairs among cancer types was still observed, the majority of pairs turned out to be cancer type-specific (S3D-E Table). Specifically, only 9,668 out of the 1,096,083 (0.9%) pooled unique [iSoMAs gene]:[iSoMAs target isoform] pairs are present in two or more cancer types; and this ratio increased slightly to 1.3% (12,997 out of 988,781) for pooled unique [iSoMAs gene]:[iSoMAs target gene] pairs (S3F-G Table). This means that despite commonality of iSoMAs genes or iSoMAs targets across cancer types, an iSoMAs gene tends to target different genes in different cancer types.

## iSoMAs genes and targets hold both biological and clinical significance in cancer

We performed a KEGG signaling pathway enrichment analysis on the top 908 iSoMAs genes that each cover at least three cancer types, defined as the 'pan-cancer' set (S2A Table). This analysis revealed significant enrichment of the 908 genes in signaling pathways associated with cell migration (e.g., Focal adhesion, ECM-receptor interaction, Protein digestion and absorption) and cell proliferation (including Growth hormone synthesis/secretion/action and PI3K-Akt signaling pathway) (Fig 4A, left; S4A Table). We also assessed the KEGG pathway enrichment profiles of the top genes ranked by the iSoMAs pipeline in each individual cancer, including those not meeting the original significance threshold p-value <1e-3 set for an iSoMAs gene. Specifically, due to the dramatically varied numbers of iSoMAs genes detected across cancer types (Table 1) and for fair comparison with the pan-cancer enrichment analysis, we also considered the top 908 (if available) mutant genes that obtained a $P_{min}$<0.05 (instead of $P_{min}$<0.001 qualifying for an iSoMAs gene) in the iSoMAs analysis (Step 2 in Fig 2A) in each cancer type. The analysis results show that, each of the top 30 significant pathways turned out to be significant in multiple cancer types in the enrichment analysis conducted for individual cancer types (Fig 4A, right; S4B Table). While a large part (n=610; 67%, S2A Table) of the 908 pan-cancer iSoMAs genes only covered three cancer types, a considerable fraction (n=22; 73%, Fig 4A, right) of the top 30 significant pan-cancer pathways proved significant in more than 3 cancer types. This implies that different member genes of the same KEGG

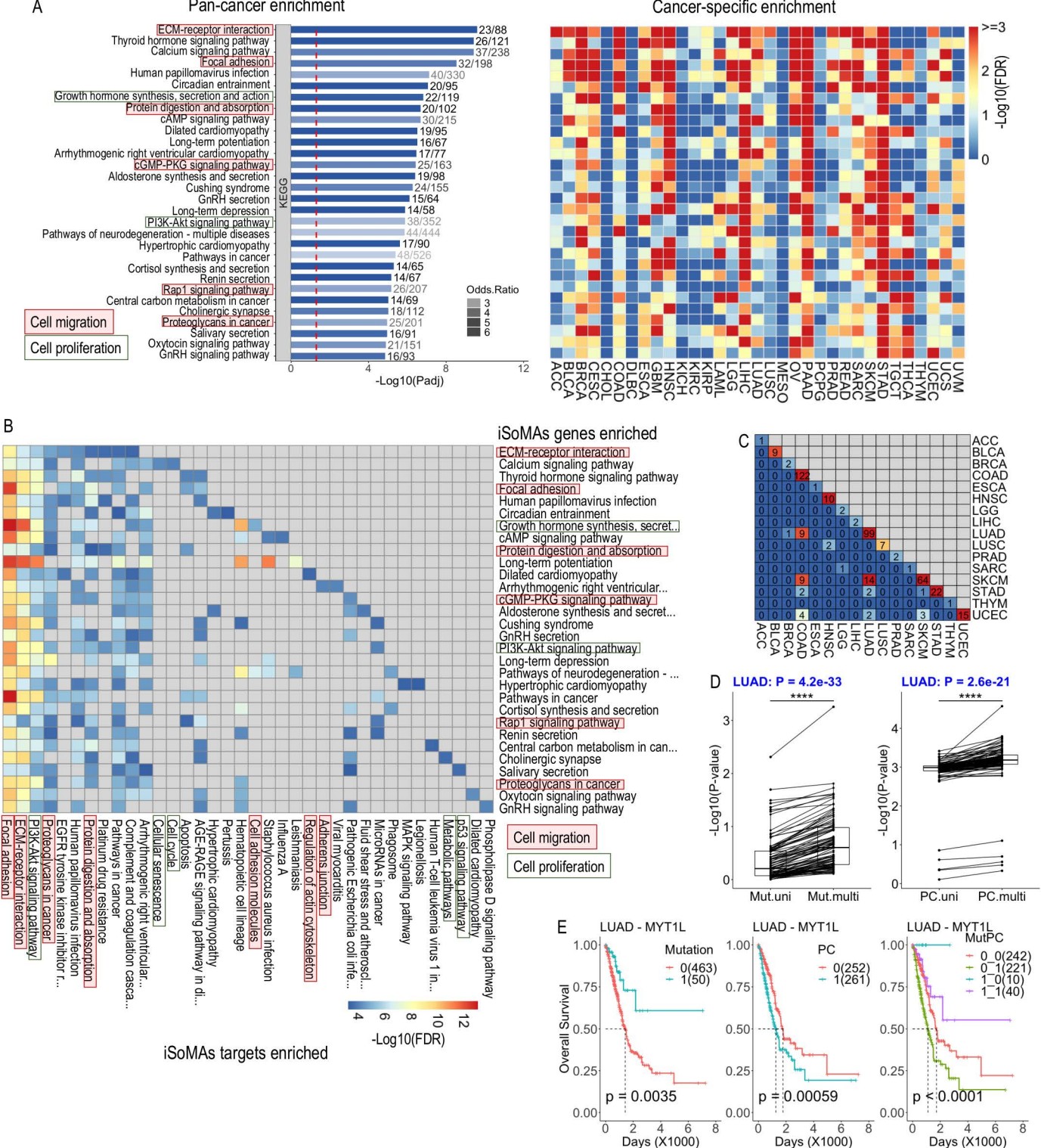

**Fig 4. Biological and clinical significance of the iSoMAs genes.** (A) Signaling pathway enrichment analysis on the top 908 iSoMAs genes (from 22,029 back-ground genes) that are tested significant in ≥3 cancer types against 343 KEGG pathways in both pan-cancer (bar plot, left) and cancer-specific (heatmap, right) manners. Only the top 30 enriched pathways are shown. In the cancer-specific enrichment analyses, gene numbers were trimmed to the top 908 (still subject to *P*<0.05 in the iSoMAs analysis) for a fair comparison. Number of genes contained in each pathway (n) and number of significant genes hitting that pathway (m) are shown on each bar (m/n). Cell migration and proliferation related pathways are marked. (B) KEGG enrichment analysis on the target isoform genes of

iSoMAs genes hitting each of the top 30 pathways shown in (A). For each pathway, the top 100 target isoform genes of each iSoMAs gene hitting that pathway were picked and pooled (with duplicates removed) to perform the enrichment analysis. Gene numbers were trimmed to the top 908 based on the frequency of presence of each gene in the pooled list. Only top 10 KEGG pathways enriched by the iSoMAs targets were used to generate the final heatmap. (C) Pairwise overlap of Additive iSoMAs genes detected across 16 cancer types. The other 17 cancer types without Additive iSoMAs genes detected are ignored. (D) Paired *t*-test compares the p-values between univariate and multivariate regression analysis for mutation (Mut.multi vs. Mut.uni, left) and its corresponding PC score (PC.multi vs. PC.uni, right) of the Additive iSoMAs genes in LUAD cancer. (E) Survival analysis on top Additive iSoMAs gene MYT1L in LUAD cancer. LUAD samples were divided into groups based on the mutation status of the gene (left), binarized PC score (middle) and combination of them (right). Mutation: 0=wildtype, 1=mutant; PC: 0=negative, 1=positive PC score. MutPC: Combination of Mutation status and PC score. Sample size of each group is indicated in parenthesis. P-values were derived from log-rank test. See also S4 Table and S4 Fig.

pathways were detected as iSoMAs genes in different cancers. This finding was further supported by a gene set enrichment analysis against the 50 Hallmark gene sets (Methods) (S4A Fig; S4C-D Table).

We further examined the biological function of the target isoforms of the top 908 pan-cancer iSoMAs genes by KEGG pathway enrichment analysis. Strikingly, these iSoMAs target genes are significantly enriched in almost the same cell migration and cell proliferation related signaling pathways as the iSoMAs genes (Fig 4B). Moreover, the iSoMAs targets show enrichment in other oncogenic signaling cascades, which likely complements the function of the iSoMAs genes. Given the observation that iSoMAs genes predominantly regulate gene isoform expression in a *trans*-acting manner, these findings suggest a synergistic mechanism where iSoMAs gene mutations and associated isoform expression alternations collectively regulate human cancer progression.

To test this synergy hypothesis, we conducted both univariate and multivariate Cox regression analysis on the two associated variables in all tumor samples of each cancer type: the iSoMAs gene mutation status and its (mathematically) associated PC score (Methods). We identified a subset of iSoMAs genes with enhanced power in distinguishing survival groups when integrated with their associated PC scores in 16 cancer types (S4B Fig), indicating a synergetic or additive effect. The additive effect refers to the higher significance level (or smaller p-value) obtained with the combination of the two variables compared to that with each variable alone in the Cox regression model. For simplicity, hereafter, we termed these special iSoMAs genes as additive iSoMAs genes (S4E Table). We observed generally negligible overlap of additive iSoMAs genes among cancer types (Fig 4C). Furthermore, a direct log-rank test confirmed this additive effect between gene mutation and PC score (binarized as 0/1 for simplicity) in representative cancer types: LUAD (Fig 4D-E) and BLCA (S4C-D Fig). Noting that the PC score is a linear combination of multiple gene isoforms, this synergistic effect further indicates a functional outcome uniting gene mutation and gene isoform expression.

## iSoMAs genes and targets are enriched for RNA binding proteins, splicing factors and transcription factors

RNA-binding proteins (RBPs) and splicing factors (SFs) are known to play a vital and direct role in shaping the gene isoform expression landscape [27]. Moreover, the expression of RBP and SF genes themselves is tightly regulated by a network of transcription factors (TFs). In this sense, the TFs are also capable of influencing gene isoform expression and theoretically can be detected as iSoMAs genes by our system. Therefore, we investigated what fraction of RBP, SF and TF genes can be identified as iSoMAs genes by our pipeline.

Generally, our pipeline detected a considerable number of RBP, SF and TF genes as iSoMAs genes that are present in one or more cancer types (Fig 5A; S5A-F Table). Particularly,

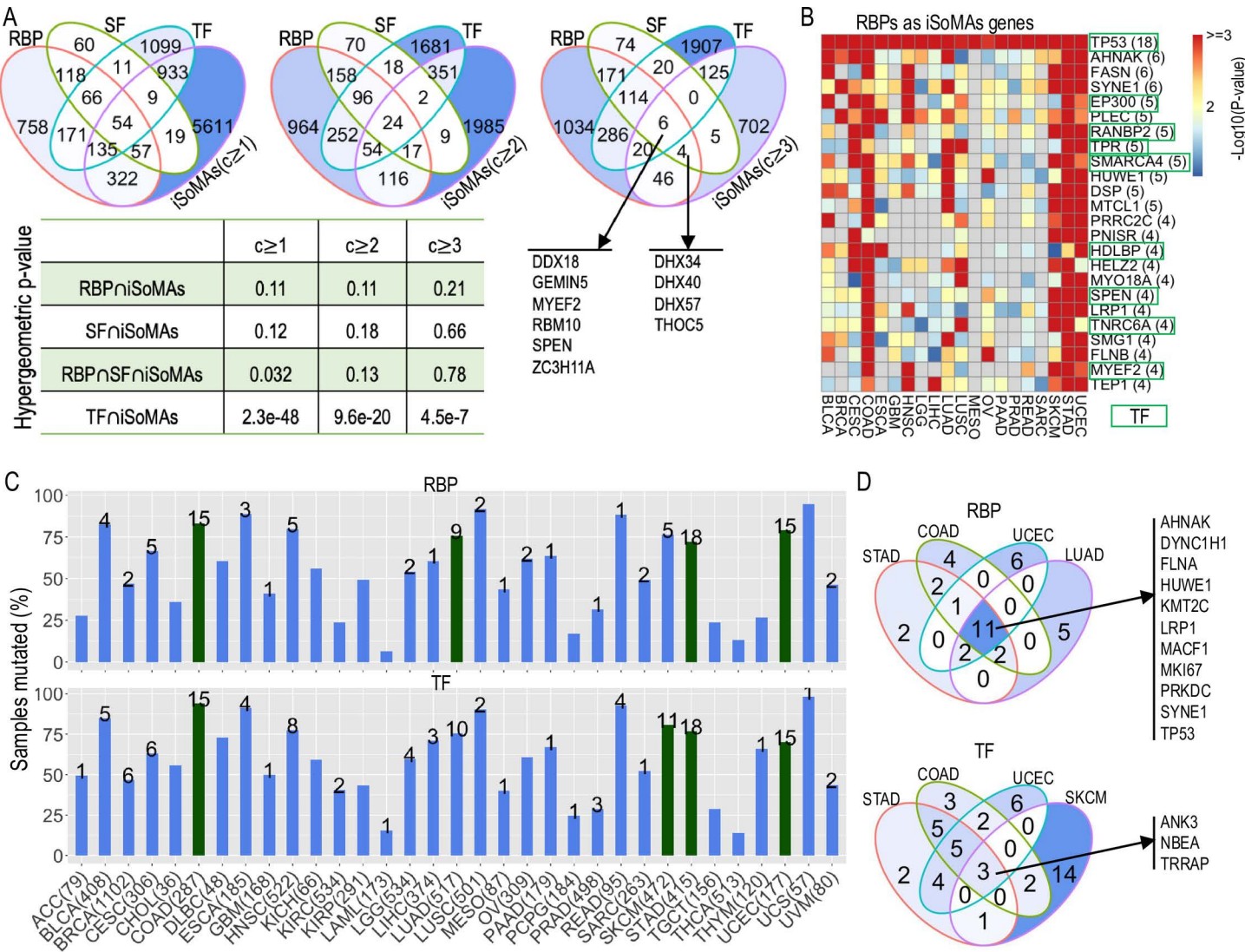

**Fig 5. RNA-binding protein (RBP), splicing factor (SF) and transcription factor (TF) genes detected as iSoMAs genes and their mutational landscape across cancer types.** (A) Venn diagram shows overlap among RNA-binding protein (RBP), splicing factor (SF), transcription factor (TF) and iSoMAs genes with increasing pan-level of cancer coverage c≥1, 2, 3. P-values were derived from hypergeometric test, with 22,029 mutant genes as background. (B) Heatmap shows the Log10-tranformed p-values of the 24 RBP genes detected as iSoMAs genes in ≥4 cancer types. Number of cancer types are indicated in parentheses. (C) Percent of tumor samples mutated in at least one of the 20 most frequently mutated RBP (upper) and TF (lower) genes in each cancer type. The number above a bar indicates the number out of the 20 RBP/TF genes detected as iSoMAs genes in the corresponding cancer type. The number of patients with gene isoform expression data available in each cancer type is also provided. The top four cancer types with most (out of the top 20) RBP and TF genes detected as iSoMAs genes are colored green. (D) Overlap of top 20 RBP (upper) or TF (lower) genes among the top four cancer types highlighted in (C). See also S5 Table and S5 Fig.

a significant part of RNA-binding SF genes (mathematically denoted RBPSF) were detected as iSoMAs genes in at least one cancer type (RBPSFiSoMAs, Hypergeometric test P=0.032); whereas the overlap between iSoMAs and RBP (RBPiSoMAs) or SF (SFiSoMAs) genes alone was not significant. We observed 10 RNA-binding SFs were detected as iSoMAs genes in three or more cancer types. Among these, four were DEAD box protein genes (*DDX18*, *DHX34*, *DHX40*, and *DHX57*), which encode RNA helicases involved in several important cellular processes, including spliceosome assembly [28]. These data imply that our iSoMAs pipeline is more likely to identify RNA-binding SFs as splicing-relevant factors (i.e., as iSoMAs genes) relative to general RBPs or SFs.

Interestingly, the TFs were consistently detected as iSoMAs genes at all pan-levels (i.e., covering at least 1-3 cancer types, respectively, Fig 5A). Particularly, TF functions are present in 6 out of the 10 widely shared RNA-binding SFs. TF functions are also included in 9 out of 24 RBP iSoMAs genes covering cancer types, including the cell cycle regulators *TP53* and *EP300* (Fig 5B; S5D, F Table). These data indicate that iSoMAs genes with TF functions often have dual roles, encompassing RBP or SF functions, which is consistent with a recent study [29].

We also checked the fraction of RBP, SF and TF genes serving as isoform targets of our detected iSoMAs genes. We observed that 34% (572/1681) of RBP genes, 32% (127/394) of SF genes and 44% (1,079/2,478) of TF genes were among the top 100 target isoforms of some iSoMAs gene in some cancer type(s) (S5G-I Table). For example, isoforms of the splicing factor gene, *HNRNPK*, represents one of the top 100 target isoforms of various numbers of iSoMAs genes across 11 cancer types (S5J Table). These results imply that: (i) multiple iSoMAs genes can target the same gene for splicing and (ii) iSoMAs genes can exert indirect effects by directly affecting splicing factors, which in turn regulate the splicing of other genes.

Consistent with their importance to gene splicing and transcription regulation, respectively, the RBP and TF genes were found to be highly frequently mutated in human cancer samples in general (Fig 5C). Specifically, a median of 56% and 60% of tumor samples bear somatic mutations in at least one of the top 20 (by mutation frequency) mutated RBP and TF genes in each of the 33 TCGA cancer types studied, respectively. In addition, among the 151 TF genes that were detected as iSoMAs genes in at least three cancer types (refer to the third Venn diagram in Fig 5A), 67 of them have shared TF targets and iSoMAs targets (S5K Table). For example, 352 (27%) of predicted iSoMAs targets of *TP53* are also previously documented TF targets of *TP53*. This percentage increased to 41% for the tumor suppressor *MZF1* [30] and 45% for proto-oncogene *FLI1* [31], respectively. These results corroborate the assertion that the iSoMAs genes can regulate gene isoform expression by acting as TFs.

The most frequently mutated RBP and TF genes include many iSoMAs genes, supporting they are associated with altered isoform expression. Specifically, we detected various numbers of iSoMAs genes from the top 20 frequently mutated RBP genes in 21 cancer types (Fig 5C, upper; Fig 5D, upper). We identified different numbers of iSoMAs genes from the top 20 frequently mutated TF genes in 26 cancer types (Fig 5C, lower; Fig 5D, lower). Interestingly, the top 20 mutated RBP and TF genes tend to mutate in a nearly mutually exclusive manner (in the sense that each sample within a cancer type only bears mutations on one or very few of the 20 genes) (S5 Fig). This exclusivity suggests a functional redundancy among the top 20 mutated RBP/TF genes in each cancer type [12], implying that they may converge to a same functional gene set related to a particular biological process [32].

## The profiles of chromatin accessibility around iSoMAs genes and their targets are coordinately disrupted

We have established that the mutation of an iSoMAs gene can be significantly associated with the isoform expression pattern of hundreds of genes (termed targets) across the genome. While the mechanisms underlying these associations are multifaceted, they primarily involve the direct or indirect impact of somatic mutations on the splicing/transcription machinery interacting with the iSoMAs gene itself or its target genes [33]. In this sense, we hypothesized that the chromatin accessibility status surrounding the iSoMAs gene itself and/or its associated target genes may undergo certain changes corresponding to the potential shift in the DNA-protein and/or RNA-protein interactions. To address this hypothesis, we resorted to the sample-matched ATAC-seq data [34] in parallel with the

TCGA DNA-seq and RNA-seq data used to build our iSoMAs pipeline. We examined the association between the chromatin accessibility profiles surrounding an iSoMAs gene and the mutation status of this iSoMAs gene. In addition, we checked the correlation between the isoform expression of the associated iSoMAs targets and their corresponding local chromatin accessibility (Methods).

Generally, the chromatin accessibility in the annotated Intron and Promoter regions showed more significant association with the iSoMAs gene mutation compared to the other four annotated regions including Distal, Exon, 3'UTR and 5'UTR (two-sided Wilcoxon rank-sum test on -Log10(HMP, or harmonic mean p-value), P=8.8e-20) (Fig 6A, upper; Fig 6B; S6A Table). This tendency is even more discernible in the parallel study detecting correlation between isoform expression and chromatin accessibility (Wilcoxon P=4.0e-100) (Fig 6A, lower; Fig 6C; S6B Table). These results indicate that both gene somatic mutation and gene isoform expression tend to have a significant association with the local chromatin accessibility status (at the level of HMP<0.05), with the significance level being specific to genomic regions. It should be noted that these data are insufficient to determine the causative relationship among these three processes (i.e., somatic mutation, isoform expression and chromatin accessibility), which necessitates further experimental investigations.

Next, we checked the combinatorial profiles of chromatin accessibility by pairing the locus of the mutant gene (iSoMAs gene) with each associated target isoform (iSoMAs target) locus with Kendall correlation analysis (Fig 6D). The correlation analysis was conducted in a pairwise manner between all ATAC-seq peak sites located in the mutant gene against those located in the target isoform, which involves multiple combinations for genes with multiple ATAC-seq peak sites. Fig 6E illustrates a specific instance of the correlation analysis conducted between *TP53* and *TPX2* in LUAD. This example highlights a notable bias specific to the genomic region type – while the correlation is significant in *TPX2*'s promoter region, it is not observed in its intron region.

Integrating all p-values obtained in the pairwise correlation analysis for each gene pair (iSoMAs gene vs. iSoMAs target), we calculated the HMP value to assess the overall correlation in chromatin accessibility between the two genes in the pair. To do so, we first calculated HMP for both iSoMAs pairs (here an iSoMAs pair refers to an iSoMAs gene and one of its associated isoform targets) and non-iSoMAs pairs (a randomly chosen pair from outside the iSoMAs pairs pool). Then we checked the frequency distribution of the HMPs of both scenarios. We found that the iSoMAs pair group invariably exhibited distinct HMP frequency distribution patterns compared to the non-iSoMAs pair group in all 33 TCGA cancer types (KS test, P<0.05), although the frequency peak of the iSoMAs group shifted to different directions in different cancers (S6A Fig). While in most cancers, the HMP frequency peak of the iSoMAs group (iSoMAs+) shifted right relative to the non-iSoMAs group (iSoMAs-), implying the iSoMAs pairs exhibited more significant correlation in chromatin accessibility compared to non-iSoMAs pairs, the HMP frequency peak of the iSoMAs group in LUAD shifts dramatically to the left (Fig 6F, left; S6A Fig). This means that in LUAD, the typically high co-occurrence of chromatin access in gene pairs was largely reduced in the somatic mutation associated isoform expression change, making a unique scenario relative to other cancer types. We further confirmed this type of peak shift was absent if the compared groups were picked from the same pools (i.e., both iSoMAs, or both non-iSoMAs pairs) (Fig 6F, middle and right; S6B-C Fig). These results corroborate that the co-profile of chromatin accessibility around iSoMAs genes and targets is largely disrupted, compared to general, randomly chosen gene pairs (Fig 6G). Therefore, integration of the ATAC-seq data brings new insights regarding the impact of somatic mutations on gene isoform expression potentially mediated by local chromatin accessibility.

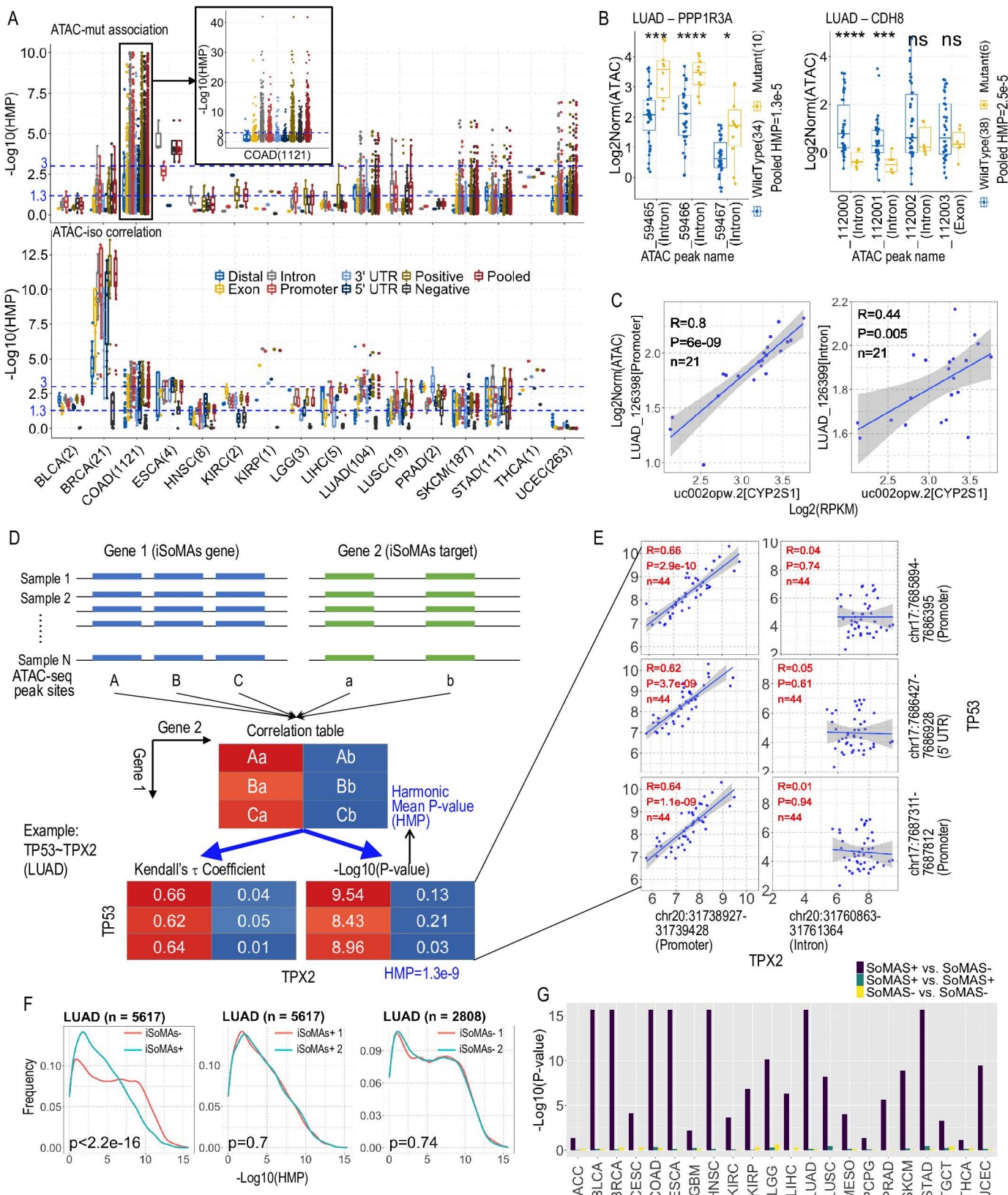

**Fig 6. Chromatin accessibility status around iSoMAs genes and targets.** (A) Boxplots show the association between gene mutation (upper, by student's *t*-test) or isoform expression (lower, by Kendall correlation analysis) and chromatin accessibility on iSoMAs genes and targets in 16 cancer types, respectively. The

number of iSoMAs genes covered in the *t*-test in each cancer type was indicated in the x-axis (iSoMAs genes without ATAC peak sites were skipped). The y-axis represents the Log10-transformed harmonic mean *p*-value (HMP) calculated for ATAC peak sites of each category as indicated. Top 100 target isoforms of each iSoMAs gene were combined to calculate the HMP in the lower panel. Positive: ATAC-seq sites yielding a positive association between gene mutation (Fold change of mutant over wildtype >0) or isoform expression (Kendall's τ coefficient >0) and chromatin accessibility; Negative: opposite to the Positive scenario; Pooled: HMP was calculated on all ATAC peak sites overlapping the gene set of interest. (B) Representative iSoMAs genes with significant association between gene mutation and chromatin accessibility in LUAD cancer. Only pooled HMP was indicated for each gene. The x-axis indicates the names of peaks (given by the ATAC-seq data with cancer type prefix LUAD_ omitted) overlapping with the iSoMAs gene in study. The y-axis Log2Norm(ATAC) value indicates the log2-transformed normalized ATAC-seq insertion counts. ****$P$<1e-4, ***$P$<1e-3, *$P$<0.05 with t-test. (C) A representative isoform targeted by multiple iSoMAs genes with significant positive correlation (Kendall) between isoform abundance and chromatin accessibility in LUAD cancer. The x-axis refers to the isoform and corresponding gene name, and the y-axis refers to the name and genomic annotation of the ATAC peaks falling in the isoform locus. (D) Schematic of workflow for assessing the correlation in chromatin accessibility between iSoMAs genes (mutant) and their targets (isoform) loci. The correlation table lists correlation profile (including Kendall's τ coefficients and p-values) of all possible combinations of ATAC-seq peak sites located in mutant (iSoMAs gene) and isoform (iSoMAs target) genes. The HMP was calculated based on the p-values in the correlation table. (E) Detailed correlation status of a representative gene pair (*TP53* vs. *TPX2*) in LUAD cancer. Shown are pairwise correlations between 3 ATAC peak sites on *TP53* and 2 sites on *TPX2*. (F) Frequency distribution of Log10-HMP values (based on pairwise Kendall correlations) for gene pairs from both iSoMAs (iSoMAs+) and non-iSoMAs (iSoMAs-) pools (left), iSoMAs+ only (middle) and iSoMAs- only (right) in LUAD. The number of gene pairs for each scenario is shown above each panel. P-values were derived from Kolmogorov-Smirnov (KS) test. (G) Summary of KS test p-values (Log10-transformed) for all 33 TCGA cancer types. The grouping strategy was same as (F). P-values <2.2e-16 were trimmed to 2.2e-16 for better visualization. See also S6 Table and S6 Fig.

## iSoMAs genes are well supported by independent cohorts and methods

While the literature lacks a comprehensive mutation – isoform expression association study (especially in pan-cancer) like iSoMAs, there indeed exist piecemeal investigations of SNV-AS associations on particular types of SNVs or AS events, which support the iSoMAs results well. For example, an integrative analysis based on multi-omics data from six TCGA cancers identified 1,030 genes harboring somatic exonic SNVs (seSNVs) that potentially disrupt the gene splicing process, including intron retention and exon skipping events [35]. iSoMAs results confirmed that 83 of these genes are associated with differential isoform expression in one or more of the six cancers (S7A Table). Strikingly, 10 out of the 14 (71.4%) confirmed TSGs with SNVs causing intron retention were identified as iSoMAs genes across different cancers.

Another study based on data from all 33 TCGA cancer types obtained 1,607 genes harboring splice-site-creating mutations (SCMs) [13], of which 103 genes were detected as iSoMAs genes in various cancers (S7B Table). Particularly, 14 out of the 16 (87.5%) highlighted genes with recurrent SCMs were detected as iSoMAs genes, often across multiple cancers (Fig 7A). A more general, independent analysis of splicing profiles using the same 33 TCGA cancers identified seven *trans*-sQTLs in six unique genes [14]. Four of the six genes (66.7%), including *SF3B1*, *IDH1*, *EGFR* and *PPP2R1A*, are discovered by iSoMAs across a varied number of cancer types (Fig 7A).

Among the genes with splicing-associated mutations, the splicing factor 3B subunit 1 (*SF3B1*) was consistently reported to be associated with differential alternative splicing of multiple protein-coding and noncoding genes in human cancers [14–16]. Particularly, AS events in *ABCC5* and *UQCC* have been confirmed as consequences of *SF3B1* mutations by qRT-PCR assays in uveal melanoma (UVM) cancer. Meanwhile, *SF3B1* mutations have been linked to patient survival, showing significant associations with improved prognosis in UVM [15] but poorer outcomes in chronic lymphocytic leukemia (CLL) cancer [36]. Although the TCGA data does not include the CLL cancer, our pipeline detected *SF3B1* as an iSoMAs gene in UVM and UCEC cancers (Fig 7B), and accurately predicted *ABCC5* and *UQCC* as its targets (ranking 150 and 88 in the unique gene list respectively) in UVM (Fig 7C). Two additional *SF3B1* targets in UVM, *ADAM12* and *GAS8* (growth arrest-specific 8), predicted by iSoMAs with marginal significance level (S7A Fig), have also been validated by the qRT-PCR assay. iSoMAs also pinpointed two novel GAS family genes, *GAS7* and *GAS6*, as *SF3B1* targets with high confidence (*GAS7* is marginally significant). The targets of *SF3B1* in UCEC are quite

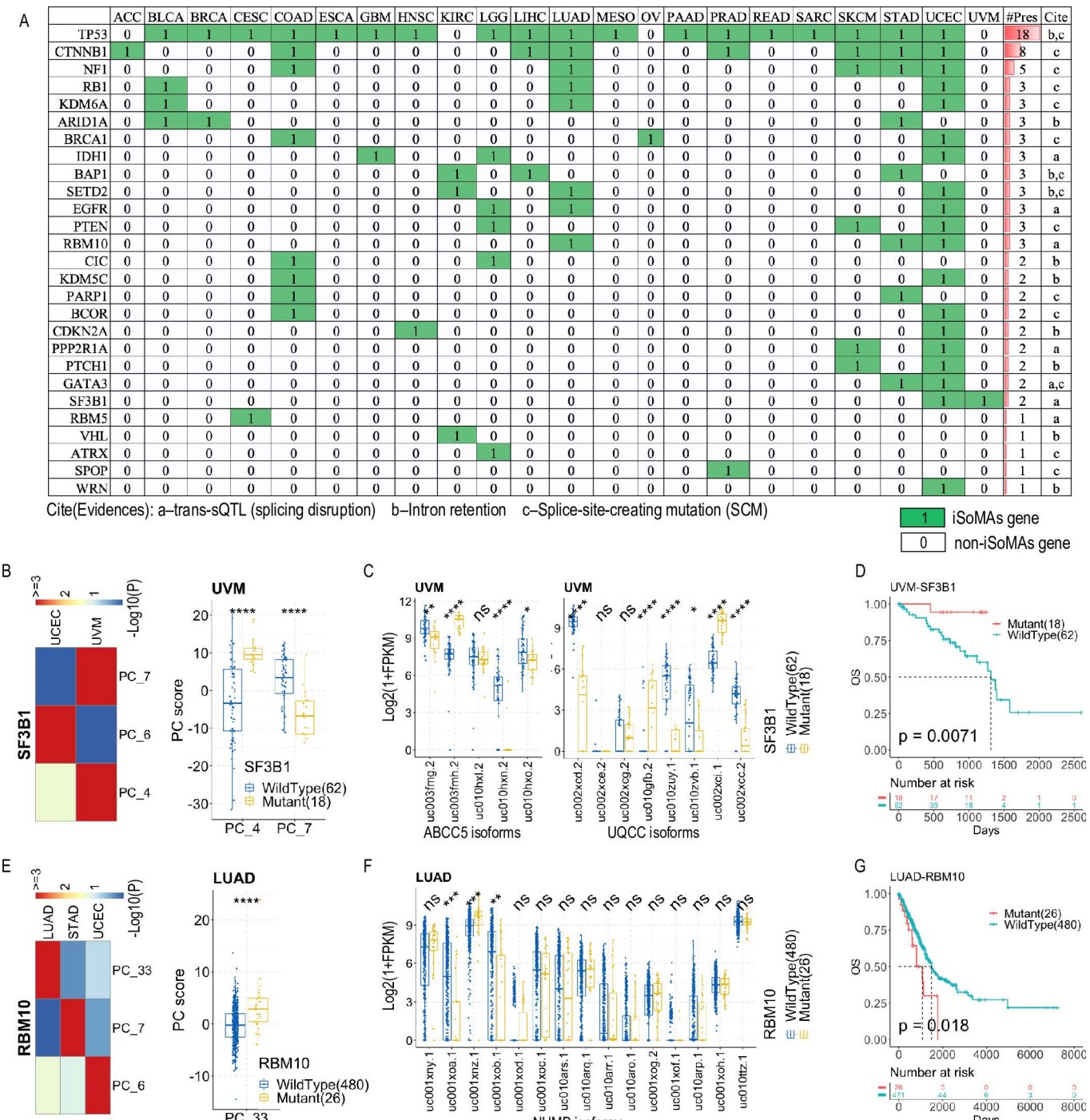

**Fig 7. iSoMAs genes are well evidenced by independent studies.** (A) A catalogue of iSoMAs genes that are explicitly supported by previous research. a: genes with *trans*-sQTLs that were reported to disrupt splicing by (Kahles et al., 2018); b: genes with SNVs causing intron retention by (Jung et al., 2015); c: genes with splice-site-creating mutations (SCMs) by (Jayasinghe et al., 2018). (B) iSoMAs analysis results for *SF3B1* in TCGA cancers. (C) Differential isoform expression analysis on *UQCC* and *ABCC5* based on *SF3B1* mutation status in UVM cancer. (D) Survival analysis based on the mutation status of *SF3B1* in UVM cancer. (E) iSoMAs analysis results for *RBM10* in TCGA cancers. (F) Differential isoform expression analysis on *NUMB* based on *RBM10* mutation status in LUAD cancer. (G) Survival analysis based on the mutation status of *RBM10* in LUAD cancer. P-values for survival analysis were derived from log-rank test; Significance levels for differential analysis were derived from Wilcoxon rank-sum test, ****$P<1e-4$, ***$P<1e-3$, **$P<0.01$, *$P<0.05$, ns: non-significant. See also S7 Table and S7 Fig.

different than those in UVM, similar to other iSoMAs genes. Particularly, *ABCC5* and *UQCC* are not top targets of *SF3B1* in UCEC, though ABCC5 does return marginal significance (S7B Fig). In addition, we corroborated that patients with mutant *SF3B1* have a much better survival than those with wildtype *SF3B1* in both UVM and UCEC cancers (Figs 7D and S7C).

Additional published research on relevant cancer genes supports our iSoMAs findings. For example, the RNA-binding motif (RBM) protein family genes, including *RBM5*, *RBM6* and *RBM10*, have been shown to regulate splicing of *NUMB* to promote growth of lung cancer cells [17]. iSoMAs detected *RBM5* as an iSoMAs gene in CESC cancer, and *RBM10* in three cancer types, including LUAD, STAD and UCEC (Fig 7E). iSoMAs also predicted that different isoforms of *NUMB* are targeted by *RBM10* in different cancers (Figs 7F and S7D, F). Consistent with the role of *RBM10* in promoting cell proliferation in lung cancer, patients with LUAD harboring *RBM10* mutations showed a significantly worse prognosis than those with wildtype RBM10 (Fig 7G). Conversely, in STAD and UCEC cancers, SNVs in *RBM10* displayed a strong protective effect against cancer (S7E, G Fig), a phenomenon that gained increasing attention recently [37]. Taking these data together, we conclude that the biological and clinical relevance of the iSoMAs genes is strongly supported by previous research employing independent methodologies and/or patient cohorts.

## Wet-lab experiments validate iSoMAs gene *TP53* in a lung cancer cell line

To validate the findings in a controlled experimental setting, we tested the role of *TP53* in the lung adenocarcinoma cell line NCI-H1975, which carries the *TP53* mutation R273H. This mutation occurs in the DNA binding domain, impairing the protein's capacity to bind the p53-responsive element [38]. Treatment of *TP53*-mutant NCI-H1975 cells with compound SCH529074 (SCH) is known to restore p53 DNA binding capacity to *TP53*-wildtype status. We confirmed this effect through the observed upregulation of a crucial downstream target of p53 (i.e., p21 [39]) in the treated (i.e., where the *TP53* function reverts to wildtype) compared to untreated (*TP53* remains mutant) group, using western blot assay (Fig 8A). The upregulation of p21 was further supported by the RNA-seq data from both TCGA-LUAD patient samples and our NCI-H1975 cell line samples (Fig 8B).

Having confirmed the action of the SCH compound at both the protein and RNA levels, we next performed differential expression analysis on the isoforms implicated in the iSoMAs analysis for *TP53* in LUAD cancer using our RNA-seq data generated from NCI-H1975 cell line. We confirmed that the top isoform targets of *TP53* in lung cancer tend to be significantly differential upon *TP53* mutation modulation (through SCH treatment) compared to the other isoforms, which is well consistent with the TCGA data (S8A Fig). This is better illustrated by the detailed differential isoform expression for two representative genes, *TPX2* and *NCAPG* (Figs 8C and S8B), which ranked No. 1 and No. 4 as *TP53* targets based on PC loadings (Fig 2H). Furthermore, we discovered that the top isoform targets of *TP53* in LUAD are enriched in the cell cycle related signaling pathways (e.g., Cell cycle, DNA replication, etc., Fig 8D). Both the *TP53* mutation status and the corresponding expression levels of *TPX2* and *NCAPG* are able to stratify LUAD cases for survival rate (Figs 8E-F and S8C). Tumor samples also have differential expression of the target isoforms compared to the normal samples, showing consistent direction of association as the survival rate (Figs 8G and S8D).

We further explored the specific effects of restoring *TP53* activity with the SCH compound on cell cycle regulation and apoptotic processes. We first tested its effect in combination with the DNA-damaging drug Adriamycin (ADR) on cell cycle. Compared to ADR alone, treatment of NCI-H1975 cells with SCH+ADR tended to retain cells in the G1 phase, even when ADR induced G2/M arrest (Figs 8H and S8E; S8 Table), demonstrating SCH's role

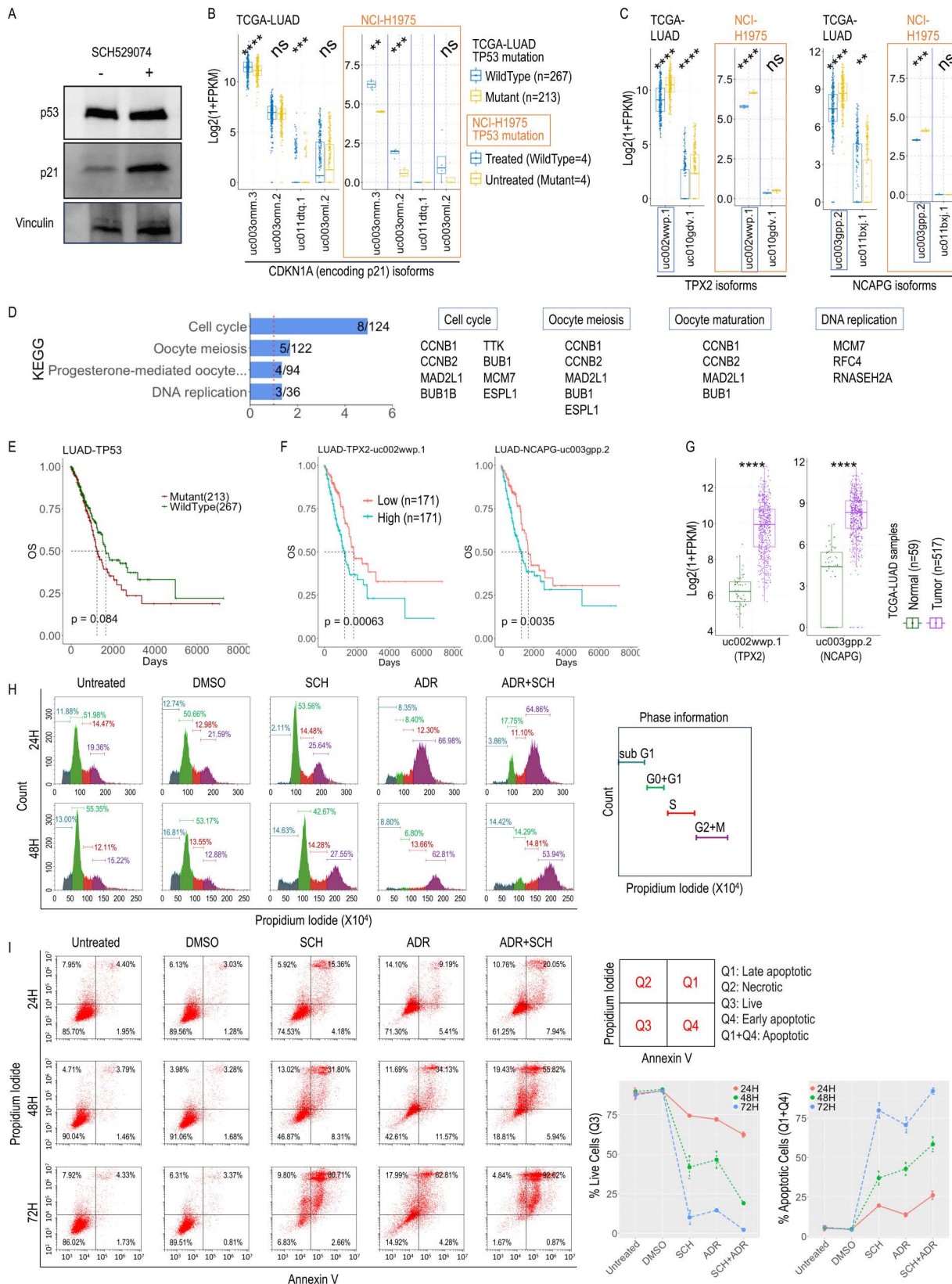

**Fig 8. Wet-lab experiments supporting iSoMAs outputs in a lung cancer cell line.** (A) Western blot generated from NCI-H1975 cell lysates. Cells were treated with 6μM SCH529074 compound (+) or mock DMSO (-) for 24 hours. Expression of p53 and p21 is shown. Vinculin

was included as a loading control. (B) Differential isoform expression for *CDKN1A* (encoding p21) upon TP53 mutation in both clinical (TCGA-LUAD) and cell line (NCI-H1975) samples as indicated. (C) Differential isoform expression for *TPX2* and *NCAPG* upon *TP53* mutation in both clinical (TCGA-LUAD) and cell line (NCI-H1975) samples as indicated. (D) Signaling pathway enrichment analysis on the top 100 isoform targets (87 unique genes) of *TP53* in LUAD against 343 KEGG pathways. Only pathways with Hypergeometric test P<0.05 are shown. (E) Survival analysis based on the mutation status of *TP53* in LUAD cancer. (F) Survival analysis based on the expression level of two representative isoform targets of *TP53* in LUAD as indicated. LUAD tumor samples were divided into 3 tiers equally based on expression level of each isoform, and only the high (n=171) and low (n=171) tiers were kept for comparison. (G) Differential expression analysis for two representative isoform targets of *TP53* between normal (n=59) and tumor (n=517) LUAD samples. (H) A representative result of the cell cycle analysis for NCI-H1975 cells under various treatment conditions at different time points as indicated. (I) Cell apoptosis analysis for NCI-H1975 cells under various treatment conditions at different time points as indicated. Percent of cells are presented as MeanSD for 2 replicates. P-values for survival analysis were derived from log-rank test; Significance levels for differential analysis were derived from Wilcoxon rank-sum test (for TCGA data) or *t*-test (for H1975 cell line data), ****$P$<1e-4, ***$P$<1e-3, **$P$<0.01, *$P$<0.05, ns: non-significant. See also S8 Table and S8 Fig.

in G0/G1-phase cell cycle regulation. Next, we examined the impact of various drug treatments on cell apoptosis using a time-coursed Annexin V and PI assay (see Methods). Compared to DMSO treatment, SCH treatment induced significantly higher rate of cell apoptosis across all three time points (Figs 8I, S8F; S8 Table). As expected, ADR induced cell death in a time-dependent manner. Interestingly, SCH induced enhanced cell death when used in combination with ADR, indicating that restoration of *TP53* aggravated the DNA-damage induced cell death in a time-dependent manner. This proof-of-principle study collectively demonstrates that our iSoMAs pipeline can detect molecular regulatory networks of high clinical relevance.

## Methods

### TCGA data acquisition and preprocessing

Gene isoform expression data, gene somatic mutation profiles, and patient clinical information for 33 cancer types were downloaded from TCGA data portal (https://portal.gdc.cancer.gov/) with the R package TCGAbiolinks [40]. Specifically, the Illumina HiSeq RNASeqV2 isoform expression level for 73,599 gene isoforms were downloaded for each individual cancer type. The isoform expression level has been normalized to FPKM values by the RSEM method [41]. These isoforms consist of 29,181 unique coding and noncoding human genes. When downloading the somatic mutation data, we chose the MUTECT2 pipeline which covers 22,029 different genes across all cancer types. We focused only on the single nucleotide variants (SNVs), which on average consists of 91.5% of all somatic mutations detected in each of the 33 TCGA cancer types (S1 Table). For clinical information, the patient survival time, quantified as 'days to death' (overall survival time) and 'days to last follow up' (censored) were extracted and converted to numeric values for subsequent analysis. Sample type (tumor vs. adjacent normal) was determined by parsing the patient barcodes, and we focused on the tumor samples in most analyses unless otherwise noted. The specific tumor stage of each tumor sample was further extracted and categorized into four large groups I-IV. The tumor purity information was derived from a published consensus measurement of purity estimation (CPE) [42].

### ATAC-seq data acquisition and preprocessing

The sample-matched ATAC-seq data for 23 out of 33 TCGA cancer types were downloaded from the NCI GDC page (https://gdc.cancer.gov/) in the form of cancer type-specific log2-transformed normalized ATAC-seq insertion count matrices (TCGA-ATAC_PanCan_Log2Norm_Counts.rds) [34]. A total of 796 individual technical replicates from 23 cancer types

were obtained, with sample size ranging from 12 in CESC to 146 in BRCA (median = 29). The number of ATAC-seq peak sites ranged from 56,112 in CESC to 215,920 in BRCA with a median of 102,550. We further extracted the genomic coordinates and annotation type (including 3' UTR, 5' UTR, Distal, Exon, Intron and Promoter) of each ATAC peak for downstream analysis.

## RBP, SF and TF genes collection and filtering

We curated a total of 1,972 RNA-binding Protein (RBP) genes from MSigDB [43] and 404 splicing factor (SF) genes from published literature [16]. Among those, 1,681 RBP and 394 SF genes overlapping with the TCGA somatic mutation data were retained in the subsequent analysis. We downloaded transcription factors (TFs) and their targets with the R data package tftargets (https://github.com/slowkow/tftargets). This dataset includes human TF information curated from six published databases: TRED, ITFP, ENCODE, Neph2012, TRRUST, Marbach2016. As previously did [44], we mapped the Entrez gene IDs into gene symbols using two R packages: annotate (1.78.0) and org.Hs.e.g.,db (3.17.0). After integration and removal of duplicates, we obtained 2,705 TFs with their target genes. We only focused on 2,478 TFs overlapping the TCGA somatic mutation data in the downstream analysis.

## Clustering of all TCGA samples based on gene isoform expression

To investigate the overall pattern of gene isoform expression across cancer types, we pooled the gene isoform expression data of all 33 TCGA cancer types together as a gene isoform-by-sample expression matrix for a clustering analysis. The original matrix includes 73,599 gene isoforms and 10,464 TCGA samples, covering 726 normal and 9,738 tumor samples. Isoforms expressed in less than 3 samples and samples expressing less than 200 isoforms were excluded for data quality control, after which we obtained a new expression matrix with 71,550 gene isoforms and 10,464 samples.

The R package Seurat [23] (4.3.0.1) was applied to implement the clustering analysis and visualization. For the clustering based on gene isoform expression, we first normalized the expression level of each sample with a LogNormalize method as $y = \log(1+x/\text{sum}(x)*10,000)$, where x is the original 71,550-long vector of isoform expression for a sample in study, and log refers to the natural logarithm. Next, we identified 3,667 most variable gene isoforms (features) with mean.var.plot (mvp) method built in the Seurat package. The variable gene isoforms behave as outliers on a 'mean variability plot', and the mvp method detects those outliers while controlling for the strong relationship between variability and average expression. Therefore, the mvp method ensures that the selected highly variable genes are not just those with high variance but also have substantial mean expression levels, which is critical for retaining biologically meaningful features. We centered and scaled the expression for each isoform by $y = (x-\text{mean}(x))/\text{sd}(x)$, where x is the vector of expression along samples for an isoform and mean and sd refer to the average and standard deviation of expression for that isoform. Then we conducted a linear dimension reduction with PCA and reduced the dimension from 3,667 (isoforms) to 50 (PCs) as default in Seurat. Finally, we implemented the clustering analysis based on the new 50-D meta-isoform expression matrix by calculating the 20-nearest neighbors and constructing the shared nearest neighbor (SNN) graph with Seurat built-in algorithms. It should be noted that while the number of PCs for dimension reduction is not biologically essential, our practice on the TCGA genomics data indicates that 50 PCs represents a good balance between computational efficiency and information retention. The choice of 50 PCs is relatively large in a general PCA application but is important for identifying most iSoMAs genes in our study.

## Clustering of all TCGA samples based on gene somatic mutation

The workflow of clustering based on gene somatic mutation is similar to that based on the gene isoform expression, while the specific configurations are a little different. To generate the feature-by-sample matrix for gene somatic mutation, we counted the number of single nucleotide variants (or SNV for short, denoted as 'SNP' in the TCGA database) consisting of 13 different variant categories for each gene in each cancer type, including 3'-Flank, 3'-UTR, 5'-Flank, 5'-UTR, IGR, Intron, Missense_Mutation, Nonsense_Mutation, RNA, Silent, Splice_Region, Splice_Site, Translation_Start_Site. Each feature is marked as a 'gene name + variant category' combination. In this way, we annotated a total of 127,009 different mutation features for all 33 cancer types. After removing features present in less than 5 samples, we eventually obtained 76,731 mutation features, and accordingly a mutation matrix of 76,731*10,178. We further adjusted the initial counts of mutations by dividing the raw mutation counts in each gene in each sample by the length of that gene. Since mutation counts are more discrete compared to isoform expression, we normalized the mutation counts of each sample by the RC (relative counts) method as y = x/sum(x)*100, i.e., feature counts for each sample are divided by the total counts for that sample and multiplied by a scale factor (100). We chose the top 2,000 variable features with the local polynomial regression model ('vst' method built in Seurat) for the subsequent linear dimension reduction (PCA, also reduced to 50-D) and clustering analysis on the mutation data.

## Comparison and visualization of different clustering results

After the clustering analysis based on the gene isoform expression and gene somatic mutation profiles, we generated the cluster residence heatmap to compare the clustering results with the cancer tissue types of all the TCGA samples, as well as the published iCluster membership [22]. We further conducted a non-linear dimension reduction analysis based on the PCA output for both gene isoform expression and gene somatic mutation data, with Uniform Manifold Approximation and Projection for Dimension Reduction (UMAP) technique [45]. TCGA samples were mapped to the 2-D UMAP space derived from gene isoform expression levels and colored in different ways, including Isoform and Mutation clusters, cancer tissue type (33 TCGA cancer types), sample type (tumor + normal), and clustering results based on other TCGA molecular data types from published work. The variation of information distance between two clustering strategies of the same objects was calculated with the vi.dist function in R package mmcluster.

## Principal component analysis (PCA) of gene isoform expression in individual cancers

The iSoMAs pipeline was applied to the tumor samples of each cancer type. The preprocessing steps for the PCA in individual cancer types, including the selection of most variable isoforms with the 'mvp' method, normalization, and scaling of the original isoform expression data, are the same as that for the pooled samples as elucidated in above sections. As illustrated in Fig 2A, the only modification is that the 'mvp' method was performed on the 59,866 transcripts derived from 15,448 multi-isoform genes instead of the original 73,599 transcripts from 29,181 gene loci. Following the preprocessing steps, we transformed the informative gene isoform expression matrix into a low-dimensional PC score matrix with PCA prior to the subsequent differential analysis. Specifically, denote the preprocessed gene isoform expression matrix and the output PC score matrix as $E$ and $\mathbf{S}$ respectively as follows,

$$\begin{bmatrix} E11 & E12\cdots & E1p \\ E21 & E22\cdots & E2p \\ \vdots & \vdots & \vdots \\ En1 & En2 & Enp \end{bmatrix} \triangleq \begin{bmatrix} E1, & E2, & \cdots, & Ep \end{bmatrix},$$

$$\begin{bmatrix} S11 & S12\cdots & S1K \\ S21 & S22\cdots & S2K \\ \vdots & \vdots & \vdots \\ Sn1 & Sn2 & SnK \end{bmatrix} \triangleq \begin{bmatrix} S1, & S2, & \cdots, & SK \end{bmatrix}$$

Where $n$, $p$ and $K$ stand for the number of samples, the number of input gene isoforms (transcripts) and the number of output PC axes (or the dimension of the new space) respectively. The PCA algorithm tries to calculate the PC loading matrix

$$L = \begin{bmatrix} L11 & L12\cdots & L1K \\ L21 & L22\cdots & L2K \\ \vdots & \vdots & \vdots \\ Lp1 & Lp2 & LpK \end{bmatrix}$$

and the PC score matrix $S$ simultaneously such that $E \times L = S$. In other words, $S$ is linearly transformed from $E$, and $L$ represents the linear transformation that stores the corresponding combination coefficients. Taking the first PC axis (PC_1) as an example, from the basic matrix multiplication operation it yields that $S_1 = \sum_{i=1}^{p} L_{i1} E_i$, which means that the PC score for all $n$ samples along PC_1 coordinate is a linear combination of the expression level of all original $p$ gene isoforms. Therefore, $S_i$ is expected to be significantly correlated (either positively or negatively) with those $E_i$'s with largest coefficients $L_{i1}$'s (either positive or negative). In this sense, the significant association between the mutation status of a gene and the PC_1 score is equivalent to the significant association between the gene mutation and those top isoforms (with largest $L_{i1}$'s). It should be noted that this logic holds across all other PC coordinates, and this makes the mathematical foundation of that the iSoMAs pipeline can identify all genes significantly associated with a mutant gene at one time by performing the differential PC score analysis instead of the direct differential gene isoform expression analysis. Throughout the work, we denote the top 100 significantly associated isoforms (and their corresponding genes depending on contexts) as the targets of the iSoMAs gene (or the gene mutation in some contexts) unless otherwise stated. It deserves noting that the choice of top 100 isoforms for downstream analysis has no biological consideration, just for mathematical convenience. This is because the iSoMAs pipeline only ranks the isoforms as being associated with the iSoMAs gene but does not determine real targeting, which subjects to validation. We tried to start with a number suitable for a meaningful downstream analysis such as enrichment profiling.

### Differential analysis and visualization

We performed three types of differential analyses in this work, including differential PC score (in iSoMAs), differential isoform expression (for RNA-seq), and differential chromatin accessibility (for ATAC-seq) analyses. In all analyses unless otherwise stated, samples were first divided into two groups based on the mutation status (wildtype vs. mutant) of a particular gene. Then a Wilcoxon rank-sum test was applied to compare the PC score or gene isoform expression between the two groups. For chromatin accessibility, $t$-test instead of Wilcoxon test was used due to smaller sample sizes. P-values of both Wilcoxon and $t$-test were provided as *$P<0.05$, **$P<0.01$, ***$P<$1e-3, and ****$P<$1e-4 unless otherwise stated. The Harmonic mean p-value (HMP) was calculated to measure the combined significance for genes with multiple ATAC-seq sites, with HMP$<0.05$ deemed significant. The ggboxplot function in the R package ggpubr was employed to visualize the differential values between different groups.

### Detection of dual-direction associations between iSoMAs gene mutation and iSoMAs target expression

A dual-direction association refers to the scenario that the mutation status of an iSoMAs gene is significantly positively associated with at least one isoform of its target gene, and at the same time, negatively associated with at least another one isoform of the same target gene, both at the $P<0.05$ level (i.e., |Log2FC|$>0$ and Pval$<0.05$) with Wilcoxon rank-sum test. Only genes representing the top 100 target isoforms of each iSoMAs gene in each cancer type were considered. For each iSoMAs gene in each cancer type, we counted the number of its target genes (falling their top 100 target isoforms) that have a dual-direction association with this iSoMAs gene and denoted it as N.Dual.sig.top100. For each iSoMAs gene in each cancer type, there can be one or more dual-direction associated genes. For each dual-direction associated gene, we first calculated the HMP (Harmonic mean p-value) of the p-values related to all isoforms having a significant association (Wilcoxon rank-sum test $P<0.05$) with the mutation status of that iSoMAs gene, then took an average over those HMP values to generate an mHMP value for that iSoMAs gene. This mHMP value is finally log10-transformed [-log10(mHMP)] to Log10mHMP to indicate the overall significance level of dual-direction association for that iSoMAs gene.

### Correlation between gene isoform expression and local chromatin accessibility

The ATAC-seq sites overlapping a specific gene isoform were first determined according to their genomic coordinates. Then the correlation between the chromatin accessibility (measured by ATAC-seq) on each site and the expression level of this gene isoform (measured by RNA-seq) was quantified by a Kendall correlation analysis. The correlation coefficients and significant levels (p-values) were provided. The Harmonic mean p-value (HMP) was calculated to measure the combined significance for genes with multiple ATAC-seq sites, with HMP$<0.05$ deemed significant.

### Calculation of inter-cancer overlap of iSoMAs genes and iSoMAs targets

The overlap of iSoMAs genes between two cancer types was measured by the Szymkiewicz–Simpson coefficient [46], which is defined as the size of intersection divided by the size of the smaller set: overlap(X, Y) = |XY|/min(|X|, |Y|). Briefly, we calculated the overlap as the number of iSoMAs genes detected in both cancer types divided by the number of iSoMAs genes detected in the cancer type with less iSoMAs genes. The overlap of iSoMAs targets

between two cancer types was determined with three steps: first, the overlapping iSoMAs genes between these two cancer types were picked; second, for each overlapping iSoMAs gene, we focused on the top 100 target isoforms of this iSoMAs gene in each cancer type and the number of overlapping targets was calculated; third, this number of overlapping targets were averaged over all the overlapping iSoMAs genes between the two cancer types, which was eventually designated as the overlap of iSoMAs targets between the involved two cancer types.

## Signaling pathway and hallmark gene set enrichment analysis

We downloaded 343 KEGG signaling pathways (March 2021) with the R package clusterProfiler [47], and 50 hallmark gene sets via the Molecular Signature Database (MSigDB) [43]. For simplicity, the member genes of all pathways/gene sets were trimmed to the background 22,029 genes covered by the TCGA somatic mutation data. In both pan-cancer and cancer-specific scenarios, we performed enrichment analysis of the genes of interest against each of the 343 KEGG signaling pathways and 50 hallmark gene sets separately with hypergeometric test. Specifically, suppose we have $q$ genes of interest, to determine their enrichment against a particular pathway (gene set) $\mathbf{S}$ with $m$ member genes (out of $N$ background genes), if $k$ out of the $q$ genes hit the pathway, we can calculate the probability of observing $k$ or more genes hitting the pathway as

$$P = \sum_{j=k}^{q} \frac{\binom{m}{j}\binom{N-m}{q-j}}{\binom{N}{q}}$$

which gives the p-value of the hypergeometric test measuring the enrichment of the genes in this pathway or gene set under investigation.

## Survival analysis

To study the clinical relevance of an iSoMAs gene and its corresponding PC score as single or combined variables, we performed both univariate and multivariate cox regression analysis on the two associated variables with the R package survivalAnalysis. The cox regression model quantifies the association between the two variables (individually or in combination) and patient survival rate. We assessed the additive effect of the two variables in comparison with each individual by comparing the p-value for the same variable derived from univariate and multivariate cox regression analysis with paired t-test. We further confirmed the additive effect by directly comparing the overall survival rate between patient groups stratified by the combination of the two variables. Specifically, tumor samples were grouped in three different ways: (1) samples were divided into iSoMAs gene wildtype and mutant groups; (2) samples were divided into positive and negative PC score groups; and (3) samples were divided into four groups based on the combination of the two variables (wildtype + positive, wildtype + negative, mutant + positive, mutant + negative). The survival difference was then quantified with log-rank test and visualized with Kaplan-Meier plot.

## Cell culture and treatment with the SCH529074 compound

NCI-H1975 cells were obtained from ATCC and grown in DMEM (Gibco, Cat. No. 105660916), 10% Fetal Bovine Serum and 100 U/mL Penicillin-Streptomycin (Life Technologies, Cat. No.15140122) at 37°C and 5% $CO_2$. Cells were initially seeded at 0.5 x $10^5$ cells

per well in a 6 well plate and grown overnight. Culture medium was replaced with medium containing 6 μM SCH529074 compound (MedChemExpress, Cat. No. HY-110088) or an equivalent volume of DMSO (mock treatment). Cells were grown for an additional 24 hours and harvested for protein or RNA.

## Protein extraction and western blot

Cell lysates were prepared using NP-40 buffer supplemented with protease and phosphatase inhibitors, and proteins were quantified via BCA assay. Samples were denatured, subjected to SDS-PAGE, and transferred to PVDF membranes. Blots were blocked, incubated with primary antibodies against Vinculin, p53, and p21, followed by HRP-linked secondary antibody.

Vinculin (E1E9V) (Cell Signaling, Cat. No. 13901) diluted 1:2000, p53 (7F5) (Cell Signaling, Cat. No. 2527) diluted 1:4000, or p21 Waf1/Cip1 (12D1) (Cell Signaling, Cat. No. 2947) diluted 1:4000, followed by anti-rabbit IgG HRP-linked Antibody (Cell Signaling, Cat. No. 7074) diluted 1:1000 in 0.1% TBS-T for 1 hour at room temperature. Detection used enhanced chemiluminescence and imaging on a BioRad GelDoc system.

## RNA-seq data generation and analysis

RNA samples of 8 NCI-H1975 cell line samples (4 untreated and 4 treated with SCH529074) were prepared for quality and quantity measurement by the Agilent 2200 TapeStation and Nanodrop. 500ng of total RNA was used in conjunction with the New England Biolabs NEBNext Ultra II Directional Poly-A RNA Library Prep Kit for Illumina (cat# E7490L and E7760). Libraries were checked by TapeStation and Qubit and sequenced on an Illumina NextSeq-550 machine. Fastq files were generated with quality controlled by FastQC (https://www.bioinformatics.babraham.ac.uk/projects/fastqc/). After that, trimmomatic-0.39 [48] was used to remove the adaptors and low-quality reads from the raw sequencing data. To be consistent with TCGA isoform expression data, the trimmed paired-end reads were aligned to the human genome hg19 using STAR-2.7.10b [49]. Aligned reads were quantified and normalized to FPKM values with RSEM based on an annotation model manually generated from the 73,599 transcripts used in TCGA (S1B Table). At last, the FPKM values were collected for downstream analysis.

## Cell cycle analysis using propidium iodide (PI)

NCI-H1975 cells were seeded into 24-well plate, on the next day when confluency reached to around 30-40%, previous culture media was replaced with fresh media with indicated concentration of drugs (DMSO, 6 uM SCH and/or 250 nM ADR). At different time points (24, 48 or 72 hours), floating as well as attached cells were collected, washed with PBS and snap-frozen on dry-ice and stored in -80$^0$C until cell cycle analysis was performed. At the time of cell cycle analysis, the cell pellets were thawed on ice and resuspended in 200μl of staining solution containing 1X buffer, 1/100x RNase, 1/200x PI (Abcam #ab287852). After 30 minutes incubation in the dark at room temperature cells were analyzed in Cytoflex FACS machine.

## Cell apoptosis analysis using Annexin V and propidium iodide (PI)

For Annexin V and PI staining, NCI-H1975 cells seeding, and drug treatment was done similarly as in the cell cycle analysis. The assay was performed immediately after collecting floating as well as attached cells together and washing them with PBS. Cell pellets were resuspended in 100ul of 1X Annexin V binding buffer followed by adding 3ul of FITC Annexin V and 2.5 μl PI (BD Pharmigen # 556547). After gently vortexing, cells tubes were incubated at RT in the

dark for 15 min followed by adding 400 µl of 1X binding buffer to each tube before analyzing by flow cytometry.

## Discussion

Somatic single nucleotide variants (SNVs) have been increasingly recognized to drive human diseases such as cancer, by inducing alternative splicing (AS) and/or altering gene expression patterns [50,51], especially at the isoform level [52]. Many efforts have been devoted to identifying SNVs that disrupt splicing (called splicing mutations) and exploring their implications in human oncogenesis. However, the previous research focused on the direct regulation of splicing by SNVs occurring in flanking exons (for *cis*-association) or a small number of known splicing factors/regulators (for *trans*-association) but ignored SNVs located in many other RNA-binding proteins (RBPs) or splicing factors (SFs), and even more upstream regulators (e.g., TFs) of the splicing machinery that function indirectly. In addition, this direct gene-by-gene association study creates a high computational burden and easily brings false positives to exhaust all SNV-AS combinations. To address these challenges, we developed a computational pipeline, called iSoMAs (iSoform expression and somatic Mutation Association), that integrates matched DNA-seq and RNA-seq data to systematically investigate the effect of somatic SNVs on isoform-level gene expression (and hence AS profiles indirectly) across the whole genome in human cancers.

We demonstrated the effectiveness of iSoMAs via benchmarking against baseline and existing methods with many examples, both computationally (Figs 2I, S2D-E and 7, S7; S2C Table) and experimentally (Figs 8 and S8). We also showed that iSoMAs is more efficient and versatile than existing methods in identification of AS-associated SNVs. Since iSoMAs implements differential analysis along a small number of PC coordinates (always 50) rather than along thousands of original transcript axes directly, it dramatically outperforms the existing models in efficiency. Furthermore, iSoMAs searches for the SNV-impacted transcripts over the whole genome simultaneously, which means that it determines both the *cis-* and *trans-* regulatory SNVs at the same time, indicating its versatility compared to previous studies. Most importantly, iSoMAs overcomes the limitation of the low mutation frequencies of most cancer genes by examining the association between an SNV and the overall genome-wide landscape of gene splicing, well reducing the false positive rate incurred by the traditional gene-by-gene association study [53].

Applying iSoMAs to 33 TCGA cancer types individually, we detected 7,140 iSoMAs genes in total, including 1,650 and 908 and iSoMAs genes that are simultaneously present in at least two and three cancer types, respectively. The iSoMAs genes included many widely known oncogenes and tumor suppressors, exemplified by the significant enrichment of iSoMAs genes in the Cancer Gene Census (CGC) genes [53]. Specifically, we found 102 out of the 733 CGC genes (14%) were present in the iSoMAs gene list covering at least three cancer types (hypergeometric test P=1.1e-27), and this number increased to 411 (56%) overlapping all the 7,140 iSoMAs genes (P=4.9e-41) (S2A Table). This verifies the hypothesis that mutations on cancer-associated genes could function via disrupting splicing and transcription processes. The iSoMAs genes also covered a considerable number of RBPs and SFs, especially those RNA-binding SFs, indicating that iSoMAs can identify direct regulators of the splicing process as well. Additionally, many TFs were identified as iSoMAs genes in various cancers, meaning that iSoMAs genes can impact gene isoform expression as a transcription regulator. Both the top 908 iSoMAs genes and their top target genes are significantly enriched in tumor growth and metastasis related processes. Some iSoMAs genes better stratify patients in terms of survival rate when in combination with expression of their target transcripts (aggregated as a PC score). These results not only provide a perfect biological explanation of why those

cross-chromosome isoforms are mathematically predicted as the top targets of certain iSoMAs genes, but also clearly demonstrate the biological and clinical significance of the identified iSoMAs genes.

Most previous studies looked for the direct association between a somatic (or germline) SNV and an explicit AS event with real split-reads support, which requires a thorough reanalysis of the raw RNA-seq data from scratch to count the split-reads, followed by an elaborate quantification of the AS event, e.g., with a percent spliced in (psi) value [14]. iSoMAs examines the association between an SNV and the expression profiles of all transcripts derived from multi-isoform genes that have been already well annotated and quantified in TCGA. iSoMAs accurately identified many genes that have been proven to directly impact gene splicing by independent methods or data sources, including splicing factors (e.g., *SF3B1*, *RBM10*) that disrupt splicing of specific genes in particular cancers through *trans*-effects [15–17], and genes with mutations that induce splice site creation nearby such as *TP53*, *GATA3*, *KDM6A*, *PTEN*, *SETD2*, *RB1* and *CTNNB1* in different cancers via *cis*-effects [13]. Collectively, we have demonstrated that mRNA abundance at the transcriptome level preserves essential information regarding the gene-splicing profiles. This enables iSoMAs to identify those splicing mutations through associations with isoform-level expression, without the need for detailed splicing analysis. This is well illustrated by the many iSoMAs genes whose mutation status is significantly associated with target isoform expression in a dual-direction manner (S2J-L Fig; S2D Table). Therefore, while iSoMAs is not designed to detect the association between SNV and AS events directly, the association between an SNV and isoform-level expression of a gene can bring many insights about the effects of the SNV on splicing of those associated genes. In this sense, SNVs on iSoMAs genes might be better interpreted as the isoform-level expression quantitative trait loci, or isoQTLs [54,55].

To better interpret this point, it's worth noting that iSoMAs focuses only on the expression of 59,866 (instead of the original 73,599) isoforms that were derived from multiple-isoform genes (Fig 2A). That is, the single-isoform genes were excluded from the very beginning. This is because single-isoform genes can be easily spotted by the traditional differential expression analysis at the gene level. Our study added value by screening differences across all isoforms, including both major and minor ones, to detect expression alterations. On the other hand, since each isoform originates from the full gene locus, every isoform, no matter major or minor, comprises a subset of all the exons. These subsets refine and specialize isoform activity versus the collective activity of all exons. In this sense, expression change of a particular isoform indicates some specialized functions of a gene are being altered. Hence, iSoMAs can evaluate the impact of a gene mutation on those specialized functions of its target genes.

The target isoforms of most iSoMAs genes are distributed across the whole genome, generally proportional to the number of transcripts located in each chromosome. This implies that the *trans*-acting associations dominate the *cis*-acting associations. A *trans*-acting association suggests that the iSoMAs gene might impact the specific isoform's expression by acting as a TF, SF, or other co-regulator of gene splicing and/or transcription. It's also possible that the target in each *trans*-association acts as an intermediate expression regulator of downstream genes. Indeed, we have shown that many RBP and SF genes ranked as the top targets of some iSoMAs genes in multiple cancer types. The splicing factor, *HNRNPK,* represents such an example. A previous study showed that *TP53* mutations modify genome-wide RNA splicing landscape by promoting *HNRNPK* expression in pancreatic cancer [18]. iSoMAs predicts *HNRNPK* as a top target of some iSoMAs genes in 11 cancer types (S5J Table). However, *TP53* was not listed among those iSoMAs genes. This might be explained by the fact that our current pipeline includes all SNVs on *TP53* whereas the literature considered two particular SNVs, i.e., the R172H/R175H on *TP53*. Indeed, when we customized our pipeline to include

only these two SNVs in PAAD cancer, we observed elevated *HNRNPK* expression in the *TP53*-mutant group compared to normal controls with marginal significance level, which shows certain consistency with the literature. Hence, our pipeline is easily calibrated to determine associations between designated genotypes and gene isoform expression patterns.

With matched ATAC-seq data we disclosed that both gene somatic mutation and associated gene isoform expression are correlated to local chromatin accessibility to some extent, especially in the intron and promoter regions (Fig 6A-C). This is comprehensible considering that the interplay between mutational process and 3-D genome organization has been reported [56]. The somatic mutation holds potential to impact the binding of chromatin-binding factors to the chromosome, thus influencing the nearby chromatin conformation and accessibility. The effect of chromatin accessibility on gene isoform expression is more straightforward and natural, in that the splicing and transcription processes need to recruit a series of factors/regulators on-site. We further revealed that the co-profile of chromatin accessibility around the iSoMAs genes and predicted targets is largely disrupted compared to randomly chosen pairs. However, the mechanisms and implications underlying this observation remain to be elucidated. One potential direction for further investigation can be extending our current multiomics (DNA-seq/RNA-seq/ATAC-seq)-based co-profile analysis from mutant-target gene pairs to multi-gene sets. One multi-gene set can be composed of different iSoMAs genes targeting the same isoform, different isoforms targeted by the same iSoMAs gene, or iSoMAs genes/iSoMAs targets that participate the same signaling pathway, etc. This extension could be straightforward because the association analysis for each constituent gene of the gene set is same with our current analysis for each member of the gene pair. The co-profile of chromatin accessibility of these functionally associated gene sets (instead of gene pairs) offers a deeper and more holistic view of the regulatory mechanisms within biological systems. It would elucidate the detailed interplay among gene somatic mutation, chromatin accessibility and isoform diversity/expression in the context of human cancer.

Some potential confounding factors need to be clarified. First, the number of iSoMAs genes detected in each cancer type has little correlation with the overall sample size (Fig 2B, upper). However, too small number of mutant samples can reduce the power of the pipeline to detect significant associations, which can be seen from the CHOL and KICH examples (S2E Table). Second, although the correlation between the numbers of iSoMAs genes and total qualifying mutant genes (mutational burden) across cancer types is statistically significant, that doesn't confound the detection of iSoMAs genes due to the overall too few iSoMAs genes detected in most cancers, i.e., no more than 10 iSoMAs genes were detected in 18 out of 33 TCGA cancer types, making the impact of mutational burden negligible. Third, we have shown that a very strict p-value correction at the PC level alone (Bonferroni method in this application, Pmin.Bonf<0.001) is sufficient to determine an iSoMAs gene at high significance level. This is because a further (secondary) p-value correction at the mutant gene level (Pmin.Bonf. FDR<0.05) couldn't filter out extra candidate iSoMAs genes (S2F Table). However, since the iSoMAs genes were detected at different significance levels, they are not equally deserving for a further investigation. We retain all iSoMAs genes at a relatively lower significance level (Pmin.Bonf.FDR<0.05) to offer the research community as many candidates as possible but recommend users to choose the top-ranked iSoMAs genes contingent on specific questions/ applications. Fourth, the gene length does not increase the likelihood for a gene to be detected as an iSoMAs gene despite a potentially higher mutation frequency among tumor samples. This is because iSoMAs compares gene isoform expression between groups with and without mutation of a particular gene. In this sense, higher mutation frequency of this gene only impacts the grouping of samples but is independent of the expression level of target isoforms

within each group. Due to this reason, we set a relatively low threshold of mutation frequency (2% of all tumor samples in each cancer type) for each mutant gene to qualify for a test.

To conclude, iSoMAs is a novel pipeline that can efficiently identify SNVs impacting gene splicing and transcription at the transcriptome level, and the pan-cancer iSoMAs genes we identified proved to bear important biological and clinical meaning.

## Supporting information

**S1 Fig. TCGA multi-omics data.** (A) Cluster residence heatmap shows the number of samples from a given Isoform cluster (BRCA Cluster) that overlaps with each of the five classic breast cancer subtypes (BRCA Subtype) including Basal, HER2E (HER2-enriched), LumA, LumB and normal-like based on PAM50 mRNA profiles (Thennavan et al, 2021) [21]. This overlap analysis includes 1,198 out of 1,215 TCGA-BRCA samples with subtype information available. True Normal: adjacent normal tissue samples; CLOW: Claudin-low samples. (B) Cluster residence heatmap shows the number of samples from a given cancer type that reside within each of the 22 (0–21) annotated Mutation clusters. (C) Somatic mutation clusters shown on the Isoform UMAP map. Numbers denote indexes of the Mutation clusters. (D) Tumor samples are shown on the Isoform UMAP map and are colored according to clustering results (Hoadley et al. 2018) [22] based on various TCGA data types as indicated. Numbers denote indexes of the clusters in each clustering scheme. (E) Tumor samples are shown on the Isoform UMAP map and are colored by tumor stage, tumor purity (Aran et al. 2015) [45] and cancer type. In (C-E), NA refers to samples absent from those used to generate the Isoform UMAP (Fig 1C).
(DOCX)

**S2 Fig. Overview and close-up look of the iSoMAs genes.** (A) Bar plot shows the number of genes tested significant along each of the 50 PC-axes in each cancer type. A gene can be counted multiple times if it tested significant along multiple PC axes. The lower panel is a zoom-in of the upper panel and plots the last 10 PCs (PC_41 – PC_50) as indicated. (B) Scaled expression levels of the top 50 positive and top 50 negative isoforms of all samples ordered by PC scores along PC_2 for *TP53* in LUAD cancer. The isoforms were ranked by the PC loading value along PC_2 axis. (C) Approximation of the PC score along PC_2 with the top isoforms as indicated by the x-labels in LUAD. The total number of variable isoforms for LUAD cancer is 3,315, as shown in the rightmost panel. (D) Direct differential isoform expression analysis on the top 50 positive (upper) and top 50 negative (lower) isoforms (ranked by PC loading) based on *TP53* mutation status. (E) Direct differential isoform expression analysis on all isoforms of the top four associated genes in (B), including positively associated *TPX2* and *CCNB1*, and negatively associated *SELENBP1* and *C16orf89*, based on *TP53* mutation status. (F) The details of differential PC score analysis along all 50 PC axes for *TP53* in LUAD cancer. TP53 was tested significant along four PC axes: PC_2, PC_4, PC_5 and PC_50 in LUAD. (G) Bar plot shows the proportion of variance explained by each of the first 100 PCs in LUAD cancer. (H) Similar to (B), heatmaps show the scaled expression levels of the top 50 positive and top 50 negative isoforms of all samples ordered by PC scores along each PC axis (PC_4, PC_5 or PC_50 as indicated) for *TP53* in LUAD cancer. The isoforms were ranked by the PC loading values along each corresponding PC axis. (I) Left: Spearman correlation between PC score (along PC_4, PC_5 or PC_50 axis as indicated) and expression level of all input 3,315 isoforms in LUAD cancer. Right: Association between TP53 mutation and expression level of all input 3,315 isoforms in LUAD cancer, measured by Wilcoxon rank-sum test. The top 50 positive and top 50 negative isoforms (ranked by PC loadings along each PC axis) are marked red and green, respectively. (J) Boxplot shows the number of iSoMAs target genes with

dual-direction association with mutation status of each iSoMAs gene detected in each cancer type. The iSoMAs gene with largest number of dual-direction associated genes is labeled for each cancer type. (K) Boxplot shows the Log10-transformed average HMP (Log10mHMP) for each iSoMAs gene in each cancer type. The iSoMAs gene with highest Log10mHMP value is labeled for each cancer type. (L) Specific examples showing dual-direction associations for representative iSoMAs genes as labeled in (G). For each labeled iSoMAs gene, the dual-direction associated gene with the largest Log10mHMP value is chosen to show. For each dual-direction associated gene, only isoforms with Wilcoxon rank-sum test P<0.05 are shown for simplicity. In (J-K), the x-axis labels the TCGA cancer types and the number of iSoMAs genes with at least one dual-direction association within their top 100 target isoforms in each cancer type. In (D-F) and (L), significance levels were derived from Wilcoxon rank-sum test, ****P<1e-4, ***P<1e-3, **P<0.01, *P<0.05, ns: not significant.
(DOCX)

**S3 Fig. More details on cis- *vs.* trans-regulation of iSoMAs genes.** (A) Detailed chromosome distribution and chromosome preference profiles of top 100 target isoforms of representative iSoMAs genes TP53 (left) and KRAS (right) across cancer types. P-values were derived from KS test for chromosome preference and are indicated above corresponding cancer types. Lhg19: the chromosome length distribution of the 24 chromosomes; Niso: number distribution of all isoforms across the 24 chromosomes. (B) Schematic of the hypergeometric test performed to determine cis-regulation of iSoMAs genes. (C) Number of cis-regulation iSoMAs genes determined at various significance levels for each cancer type as indicated. The cancer types, together with the number of iSoMAs genes with *P*>0.999 (potentially trans-regulation iSoMAs genes) and the number of all iSoMAs genes detected in each cancer type are indicated in the x-axis.
(DOCX)

**S4 Fig. More results on biological and clinical significance of the iSoMAs genes.** (A) Signaling pathway enrichment analysis on the top 908 iSoMAs genes that tested significant in ≥3 cancer types against 50 MSigDB Hallmark gene sets in both pan-cancer (bar plot, left) and cancer-specific (heatmap, right) manners. Gene numbers were trimmed to the top 908 (still subject to *P*<0.05 in the iSoMAs analysis) in the cancer-specific enrichment analyses for a fair comparison. Number of genes in each gene set (n) and number of iSoMAs genes hitting that gene set (m) are shown on each bar (m/n). Cell migration and proliferation related gene sets are marked. (B) Univariate and multivariate Cox regression analysis on the iSoMAs gene mutation and its corresponding PC score across pan-cancer. iSoMAs: number of all iSoMAs genes detected in each cancer type; Test: number of cancer-specific iSoMAs genes overlapping with the 984 pan-cancer iSoMAs genes; Significant: number of iSoMAs genes tested significant in either univariate or multivariate Cox regression analysis in each cancer type; Additive: number of iSoMAs genes for which the multivariate analysis yielded higher significance level (or smaller p-value) compared to the univariate regression analysis. (C) Paired t-test compares the p-values between univariate and multivariate regression analysis for mutation (left) and its corresponding PC score (right) of the Additive iSoMAs genes in BLCA cancer. (D) Survival analysis based on the Additive iSoMAs gene PRKDC in BLCA cancer. BLCA samples were divided into groups based on the mutation status of the gene (left), the sign of its corresponding PC score (middle) and combination of them (right). Mutation: 0=wildtype, 1=mutant; PC: 0=negative, 1=positive PC score. MutPC: Combination Mutation_PC score. Sample size of each group is indicated. P-values were derived from log-rank test.
(DOCX)

**S5 Fig. Mutational landscape of the top RBP and TF genes in representative cancer types.** Shown are the top 20 frequently mutated RBP (A) and TF (B) genes in each of the four cancer types shown in Fig 5D. Bars on the top indicate the number of mutations (Log10-transformed) detected in the top 20 RBP genes in each tumor sample. Bars on the right indicate the minimum p-value (Log10-transformed) along the 50 PCs obtained in the iSoMAs analysis for each gene detected as iSoMAs gene in corresponding cancer type.
(DOCX)

**S6 Fig. More details on co-profiles of chromatin accessibility surrounding iSoMAs gene-target pairs.** (A) Frequency distribution of Log10-HMP values for gene pairs from both iSoMAs (iSoMAs+) and non-iSoMAs (iSoMAs-) pools in 33 cancer types. (B) Frequency distribution of Log10-HMP values for gene pairs from iSoMAs+ pool only in 33 cancer types. (C) Frequency distribution of Log10-HMP values for gene pairs from iSoMAs- pool only in 33 cancer types. The number of gene pairs for each scenario is shown above each panel. The p-values were derived from Kolmogorov-Smirnov (KS) test.
(DOCX)

**S7 Fig. Additional validation results for iSoMAs genes SF3B1, RBM10 and TP53.** (A) Differential isoform expression analysis on four genes as indicated, based on SF3B1 mutation status in UVM cancer. (B) Differential isoform expression analysis on ABCC5 and UQCC based on SF3B1 mutation status in UCEC cancer. (C) Survival analysis based on the mutation status of SF3B1 in UCEC cancer. (D) Differential isoform expression analysis on NUMB based on RBM10 mutation status in STAD cancer. (E) Survival analysis based on the mutation status of RBM10 in STAD cancer. (F) Differential isoform expression analysis on NUMB based on RBM10 mutation status in UCEC cancer. (G) Survival analysis based on the mutation status of RBM10 in UCEC cancer. P-values for survival analysis were derived from log-rank test; Significance levels for differential analysis were derived from Wilcoxon rank-sum test, ****$P$<1e-4, ***$P$<1e-3, **$P$<0.01, *$P$<0.05, ns: non-significant.
(DOCX)

**S8 Fig. Additional details for wet-lab experiments in NCI-H1975 lung cancer cell line.** (A) Differential expression analysis on NCI-H1975 lung cancer cell line data for top (upper panel) and bottom (middle panel) 25 isoform targets of TP53 ranked by feature loadings derived from iSoMAs analysis on TCGA-LUAD, as well as randomly chosen 25 isoforms (lower panel) from outside the iSoMAs input. TP53 Mutant group corresponds to the original H1975 lung cancer cell line samples (n=4), TP53 WildType group refers to the samples treated with SCH529074 compound (n=4). (B) Gene structure (hg19) of specific isoforms of TPX2 and NCAPG in the TCGA isoform expression data. (C) Survival analysis based on the expression level of representative positive (left panel) and negative (right panel) isoform targets of TP53 in LUAD. LUAD tumor samples were divided into 3 tiers equally based on expression level of each isoform, and only the high (n=171) and low (n=171) tiers were kept for comparison. (D) Differential expression analysis for representative positive (left panel) and negative (right panel) isoform targets of TP53 between normal (n=59) and tumor (n=517) LUAD samples. (E) Cell cycle analysis results for NCI-H1975 cells under various treatment conditions at different time points as indicated. (F) Cell apoptosis analysis for NCI-H1975 cells under various treatment conditions at different time points as indicated. P-values for survival analysis were derived from log-rank test; Significance levels for differential analysis were derived from Wilcoxon rank-sum test (for TCGA data) or $t$-test (for H1975 cell line data), ****P<1e-4, ***P<1e-3, **P<0.01, *P<0.05, ns: non-significant.
(DOCX)

**S1 Table.  TCGA multi-omics data used in the study.**
(XLSX)

**S2 Table.  iSoMAs genes detected across pan-cancer.**
(XLSX)

**S3 Table.  iSoMAs pairs present in pan-cancer.**
(XLSX)

**S4 Table.  Biological and clinical relevance of iSoMAs genes.**
(XLSX)

**S5 Table.  RBP, SF and TF genes detected as iSoMAs genes.**
(XLSX)

**S6 Table.  iSoMAs and ATAC association profiles.**
(XLSX)

**S7 Table.  Independent evidence for iSoMAs genes.**
(XLSX)

**S8 Table.  Cell line experiments supporting iSoMAs predictions.**
(XLSX)

## Acknowledgments

We thank all members of the Elnitski Lab for valuable discussions. We appreciate Stacie Anderson and Martha Kirby from the NHGRI Flow Cytometry Core and Bayu Sisay from the NHGRI Microarrays and Single-Cell Genomics Core for their technical support in our wet-lab experiments. This work utilized the computational resources of the NIH HPC Biowulf cluster (http://hpc.nih.gov).

## Author contributions

**Conceptualization:** Hua Tan, Laura Elnitski.

**Formal analysis:** Hua Tan.

**Funding acquisition:** Laura Elnitski.

**Investigation:** Hua Tan, Valer Gotea, Sushil K Jaiswal, Nancy E Seidel, Sara R Bang-Christensen.

**Methodology:** Hua Tan, Valer Gotea, Laura Elnitski.

**Software:** Hua Tan.

**Supervision:** Laura Elnitski.

**Validation:** Hua Tan, Sushil K Jaiswal, Nancy E Seidel.

**Visualization:** Hua Tan, Sushil K Jaiswal, Laura Elnitski.

**Writing – original draft:** Hua Tan.

**Writing – review & editing:** Hua Tan, Valer Gotea, Sushil K Jaiswal, Nancy E Seidel, David O Holland, Kevin Fedkenheuer, Abdel G Elkahloun, Sara R Bang-Christensen, Laura Elnitski.

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
