## [Decision Letter · Decision Letter 0]

10 Jul 2024

Dear Dr. Elnitski,

Thank you very much for submitting your manuscript "iSoMAs: Finding isoform expression and somatic mutation associations in human cancers" for consideration at PLOS Computational Biology.

As with all papers reviewed by the journal, your manuscript was reviewed by members of the editorial board and by several independent reviewers. In light of the reviews (below this email), we would like to invite the resubmission of a significantly-revised version that takes into account the reviewers' comments.

We cannot make any decision about publication until we have seen the revised manuscript and your response to the reviewers' comments. Your revised manuscript is also likely to be sent to reviewers for further evaluation. In particular, the revision will need to address the major concerns related to the comparison with baseline and state-of-the-art methods, as well as related to the interpretability of the results.

Sincerely,

Simone Zaccaria

Academic Editor

PLOS Computational Biology

Alison Marsden

Section Editor

PLOS Computational Biology

Reviewer's Responses to Questions

**Comments to the Authors:**

Reviewer #1: Tan et al. demonstrate iSoMAs, a novel computational pipeline based on principal component analysis (PCA), to investigate the association between gene somatic mutations and isoform expression in TCGA cancer data. The pipeline involved two primary steps:

1. The high-dimensional isoform expression matrix was refined using the mean.var.plot method from Seurat to retain only the most variable isoforms. This high-variable isoform matrix was then further compressed to 50 dimensions via PCA. Each principal component (PC) represented a linear combination of isoforms, referred to as meta-isoforms.

2. A Wilcoxon rank-sum test was performed separately for each of the 50 pcs according to the individual mutation status of each SNV. The Bonferroni correction procedure was then applied at the PC level for multiple testing corrections.

The authors applied iSoMAs to 33 TCGA cancer types and aggregated all 7,140 significant genes into a unique iSoMAs gene set. There are 908 iSoMAs genes identified in three or more cancer types, while 4,582 genes are exclusive to one cancer type. Statistical analyses revealed that iSoMAs target isoforms showed no preference between cis- and trans-acting mechanisms, involving well-known oncogenes, tumor suppressors, RNA-binding proteins, and splicing factors. Additionally, the authors also performed wet-lab experiments to validate the top iSoMAs gene TP53.

The author has proposed an innovative idea and developed an efficient iSoMAs algorithm capable of capturing cross-chromosomal correlations. However, further analysis and explanation are needed regarding the robustness, interpretability, and statistical analysis of the multiple testing correction issue. Thus, our suggestion is "major revision".

Major comments:

1. The authors trimmed an approximately 70,000-dimensional whole-genome isoform expression matrix to 3,667 dimensions using the mean.var.plot method from Seurat and then further reduced the dimensions to 50 using PCA, instead of directly associating SNV grouping of a specific gene with the expression of certain isoforms. The association is between mutated genes and meta-isoforms (linear combinations of isoforms). Several questions need to be answered here:

a. Seurat is designed for single-cell expression data where the read counts could be sparser and noisier, is there a specific reason using it for the TCGA individual level bulk seq data? And the default method for identifying high variable genes in Seurat is vst. Why choose the mean.var.plot method to select highly variable isoforms? How was the number 3,667 determined? Are iSoMAs genes robust for the method of selecting highly variable isoforms?

b. Why was the number of dimensions reduced to 50PCs? What if fewer PCs were selected, such as 10 or 20, or if the dimensions were increased to 100? Beyond the number of iSoMAs genes, we think it is also crucial to assess whether the same iSoMAs genes would consistently appear and how their PC loadings would change. Since the current target isoforms seem to be distributed genome-wide without chromosomal preference, we are curious whether the target isoforms (and their PC loading weights) for the same iSoMAs gene would vary drastically with different parameter settings.

c. The current iSoMAs analysis involves a nested, two-layer statistical test, like QTL analysis. In certain cancer types, the number of tests for the first layer, the mutant genes layer, can reach up to 19,162 (UCEC). For QTL calculations, a popular approach is to apply the Benjamini-Hochberg or Storey q-value method for the first layer of FDR control at the gene level, followed by permutation-based multiple corrections at the SNP level. The authors addressed the issue of multiple testing corrections. However, while strict correction was applied at the PC level using the Bonferroni method, the gene-level multiple testing correction seems absent.

2. The iSoMAs process is efficient but lacks interpretability.

a. PCA is a linear dimensionality reduction method. It does not generate meta-isoforms based on prior knowledge like gene/isoform interactions but just maximizes variance retention among samples and produces orthogonal PCs. As a result, biologically implausible isoform combinations may arise, and this issue has not been thoroughly discussed. What is the actual biological explanation of cross-chromosome meta-isoforms? Why are the first 100 isoforms considered targets of mutant genes? Do these isoforms interact with each other? Can PCA capture isoforms of the gene itself or its interacting genes into a single PC and assign them large weights?

b. When using PCA for dimensionality reduction, we do not treat each PC equivalently because they explain varying proportions of the total variance in the data. The first few PCs may explain up to 90% of the variance. However, iSoMAs does not differentiate between PCs. Any gene significant in one single PC will be considered an iSoMAs gene. Compared with iSoMAs genes identified with PC1, do iSoMAs genes identified with PC 40, PC 50, etc., truly exist and show a strong association with a specific isoform? We would like to see further analysis.

c. The lack of interpretability is also reflected in the absence of a preference for cis/trans regulation of iSoMAs genes. In traditional QTL analysis, cis QTLs are usually significantly stronger than trans QTLs and are easier to identify. This is not only because cis effects are methodologically easier to detect, but also biologically more plausible. The physical proximity of a variant to a specific gene/isoform is naturally more likely to affect the expression of that gene/isoform. While iSoMAs can capture both cis and trans effects, it is surprising that iSoMAs do not capture relatively more cis effects, which should be more significant and easier to detect biologically. We hope to see more analysis comparing the iSoMAs with the more classic approach: directly associating SNV grouping of a specific gene with the expression of the target isoforms.

Minor comments

1. Some of the figure labels are incorrect, such as in Figure 2a:

- Multi-isoform: n! 59,866

- Original: n! 73,599

These annotations should be corrected.

2. Wording

In the wet lab validation part, it is well established that TP53 is a cancer-related gene. More discussion directly related to iSoMAs may be needed and some of the current wording and expressions are a bit unclear.

For example, why do we need to check the expression of TPX2, and NCAPG isoforms? Are these isoforms the top target isoforms of TP53? How do they rank among the top 100 targets? It’s further discussed in the supplement figure S8A but not clearly explained in the main text.

Reviewer #2: The authors proposed a method to detect mutation-isoform expression association. It starts by performing PCA on the isoform expression matrix, and then conducts a differential PC score analysis by investigating the association between gene mutation status and the PC score of isoform expression. For each mutant gene, the top ranked isoforms (those with large PC loadings of the most significantly associated PC score) is selected for downstream analysis. Overall, the methodological contribution is limited, parameter settings of the method seem to be arbitrary, and the advantage of the method is not compared with state-of-the art methods. My concerns are provided below.

1, The method aims to detect mutation-isoform expression association by performing differential PC score (i.e. a weighted linear combination of original isoforms) analysis of isoform expression between mutant genes and the wild type. (1) For each gene, the authors consider only the most significantly associated PC score, which obviously leads to information loss since the remaining PCs may also be associated with the gene. How the isoforms corresponding to the other PCs are associated with the gene are not assessed. (2) For each gene, the top ranked 100 isoforms associated with large PC loadings are deemed as target isoforms. The significance of the association between individual isoforms and the gene is not statistically evaluated. (3) In PCA, the direction of the PC score only represents the direction with maximal variance; therefore, the direction of PC score does not necessary most correlate with the gene status (mutant or wild type); this might compromise the statistical power of the method to detect mutant-isoform expression association.

2, A simple and necessary baseline is to perform differential isoform expression between mutant and wild type genes. Benchmarking against this baseline and existing methods were not conducted.

3, The authors identified more than 2,000 iSoMAs genes in STAD and UCEC cancers which are known for hypermutability, but none in CHOL and KICH cancers. Can significant associations be detected in CHOL and KICH cancers using conventional differential isoform expression (not using PCA) or other methods?

4, Arbitrary parameter settings. For example, why only one (i.e. the most significantly associated) PC score is considered? Why is the number of selected isoforms for the considered PC is set to 100? These settings do not have experimental support.

Reviewer #3: The article “iSoMAs: Finding isoform expression and somatic mutation associations in human cancers” by Tan et al describes the development of a novel computational pipeline called iSoMAs (iSoform expression and somatic Mutation Association). This pipeline integrates DNA-seq and RNA-seq data to identify associations between somatic SNVs and isoform expression profiles in different cancer types. Using PCA, iSoMAs assesses the association between SNVs and the overall transcriptome expression pattern, rather than direct one-to-one associations. Applied to 33 cancer types from the TCGA database, iSoMAs identified 908 genes whose mutations are linked to altered isoform production in multiple cancer types, as well as additional genes specific to one or two cancer types. These genes include known oncogenes, tumor suppressors, and various regulatory proteins, highlighting their biological and clinical importance.

Authors validated the findings through independent studies, including the reversal of TP53 mutation effects in lung cancer cells. iSoMAs is noted for its computational efficiency and comprehensive modeling of cancer cell systems biology, making it the first PCA-based method to explore the association between somatic SNVs and transcriptome-wide isoform expression across various cancers.

The article is well-written, presenting its methodologies and findings with clarity and precision. The analyses are very comprehensive, offering a thorough exploration of the associations between somatic mutations and isoform expression profiles across multiple cancer types. I only have minor comments/suggestions for improvement:

Explain acronyms such FPKM, RSEM as soon as they appear.

Figure 8H and 8I – text is too small to read

Given the potential of iSoMA to indicate e.g., chromatic accessibility, while pairwise gene interactions provide valuable information, multiple gene interaction analysis offers a deeper and more holistic view of the regulatory mechanisms within biological systems. Perhaps, adding a discussion paragraph on how these could be done and help identify and understand the mechanisms underlying disrupted chromatin accessibility would be helpful.

**Have the authors made all data and (if applicable) computational code underlying the findings in their manuscript fully available?**

Reviewer #1: Yes

Reviewer #2: None

Reviewer #3: None

PLOS authors have the option to publish the peer review history of their article (what does this mean? ). If published, this will include your full peer review and any attached files.

**Do you want your identity to be public for this peer review?** For information about this choice, including consent withdrawal, please see our Privacy Policy .

Reviewer #1: No

Reviewer #2: No

Reviewer #3: No
---

## [Decision Letter · Decision Letter 1]

18 Oct 2024

Dear Dr. Elnitski,

Thank you very much for submitting your manuscript "iSoMAs: Finding isoform expression and somatic mutation associations in human cancers" for consideration at PLOS Computational Biology.

As with all papers reviewed by the journal, your manuscript was reviewed by members of the editorial board and by several independent reviewers. In light of the reviews (below this email), we would like to invite the resubmission of a significantly-revised version that takes into account the reviewers' comments.

We cannot make any decision about publication until we have seen the revised manuscript and your response to the reviewers' comments. Your revised manuscript is also likely to be sent to reviewers for further evaluation.

Sincerely,

Simone Zaccaria

Academic Editor

PLOS Computational Biology

Marc Birtwistle

Section Editor

PLOS Computational Biology

Reviewer's Responses to Questions

**Comments to the Authors:**

Reviewer #1: This revision represents a significant improvement upon the original submission, and we appreciate the effort the authors have invested in thoroughly examining the behavior of their method. Below are some specific comments and questions:

1.a

This is sufficiently addressed.

1.b

The authors mentioned that TDRD6, the most significant potential iSoMAs gene identified with PCs behind PC_50 in LUAD, turned out to be in-significant upon validation with isoform expression levels.

We have two inquiries regarding this:

1. In the lower-left corner of Figure R3, two isoforms show significant differences. Why is it stated that there is no significance at the isoform level? Is this due to the majority of isoforms being non-significant?

2. For the iSoMAs genes identified within the PC_01 to PC_50 range, has isoform-level significance also been validated using the same method, especially for the candidate genes identified in the later PCs (such as PC_41 to PC_50)?

1.c

The authors stated the gene-level multiple testing correction was not neglected but investigated in a subtle way. We would appreciate further clarification on this.

1. The authors indicated that the first step of iSoMAs involves obtaining the minimum p-value across the 50 PCs (Pmin) for each test gene. If the first significant PC (i.e., the PC with the smallest index among all significant PCs) was selected for downstream analysis, should the p-value of this first significant PC (Pfirst), instead of the Pmin be used for multiple testing correction? Given that different PCs represent different meta-isoforms, we suppose these two cannot be conflated.

2. For the hierarchical two-layer correction procedure, the multiple hypothesis tests for candidate genes were controlled (Step 2, global correction) based on the multiple testing-adjusted statistics (Step 1, local correction) of each gene's best association, rather than the initial p-values. For iSoMAs, if the local correction is performed using the Bonferroni procedure, the second-level FDR should be applied to the adjusted p-values (Padj-Bonferroni) rather than the initial minimum p-values (Pmin). However, it appears that the current FDR05 set was generated with Pmin, which makes it a pooled single-layer correction procedure. We are concerned about this because the pooled FDR methods (even the strict Bonferroni procedure) often fail to control the FDR of target genes.

2.a

This is sufficiently addressed.

2.b

This is sufficiently addressed.

2.c

This is sufficiently addressed.

Reviewer #2: My previous comments were not well addressed, which are described below.

(1) Regarding point 2 of my first comment (“…The significance of the association between individual isoforms and the gene is not statistically evaluated”), the authors provide only an example using the TP53 gene (Figure S2D-E, added Figure 2I, Table S2C and Figure R10). A thorough benchmarking on the different types of cancers considered in this work is necessary (but not performed), because this is key to assess the advantage of IsoMA over conventional differential isoform expression analysis. Without such benchmarking, it is not sufficient to demonstrate of the power/contribution of ISoMA (which associations are uniquely identified by ISoMA? Which are unique to conventional methods? Which are shared by both?). Actually, taking Figure R12 in the response letter as an example, conventional methods do identify significantly differentially expressed isoforms that ISoMAs cannot detect.

(2) Regarding point 3 of my first comment (“the direction of the PC score only represents the direction with maximal variance; therefore, the direction of PC score does not necessary most correlate with the gene status (mutant or wild type)”), here by “direction”, I mean the projection direction of the data that explain a maximal amount of variance (a detailed explanation of direction can be found elsewhere, such as https://builtin.com/data-science/step-step-explanation-principal-component-analysis), but not the positive/negative sign.

Reviewer #3: The authors addressed my comments therefore I approve the revisions.

**Have the authors made all data and (if applicable) computational code underlying the findings in their manuscript fully available?**

Reviewer #1: Yes

Reviewer #2: None

Reviewer #3: Yes

PLOS authors have the option to publish the peer review history of their article (what does this mean? ). If published, this will include your full peer review and any attached files.

**Do you want your identity to be public for this peer review?** For information about this choice, including consent withdrawal, please see our Privacy Policy .

Reviewer #1: No

Reviewer #2: No

Reviewer #3: No
---

## [Decision Letter · Decision Letter 2]

3 Feb 2025

Dear Dr. Elnitski,

We are pleased to inform you that your manuscript 'iSoMAs: Finding isoform expression and somatic mutation associations in human cancers' has been provisionally accepted for publication in PLOS Computational Biology.

Best regards,

Simone Zaccaria

Academic Editor

PLOS Computational Biology

Alison Marsden

Academic Editor

PLOS Computational Biology

Reviewer's Responses to Questions

**Comments to the Authors:**

Reviewer #1: The authors have addressed all my comments and questions.

Reviewer #2: My previous concerns were addressed through experiments and clarification.

**Have the authors made all data and (if applicable) computational code underlying the findings in their manuscript fully available?**

Reviewer #1: Yes

Reviewer #2: Yes

PLOS authors have the option to publish the peer review history of their article (what does this mean? ). If published, this will include your full peer review and any attached files.

**Do you want your identity to be public for this peer review?** For information about this choice, including consent withdrawal, please see our Privacy Policy .

Reviewer #1: No

Reviewer #2: No

---

## [Editor Report · Acceptance letter]

PCOMPBIOL-D-24-00829R2

iSoMAs: Finding isoform expression and somatic mutation associations in human cancers

Dear Dr Elnitski,

I am pleased to inform you that your manuscript has been formally accepted for publication in PLOS Computational Biology. Your manuscript is now with our production department and you will be notified of the publication date in due course.

With kind regards,

Zsofia Freund
